# DeepReduce: A Sparse-tensor Communication Framework for Federated Deep Learning

**Hang Xu**
KAUST
hang.xu@kaust.edu.sa

**Kelly Kostopoulou**
Columbia University
kelkost@cs.columbia.edu

**Aritra Dutta**
KAUST
aritra.dutta@kaust.edu.sa

**Xin Li**
University of Central Florida
xin.li@ucf.edu

**Alexandros Ntoulas**
NKUA
antoulas@di.uoa.gr

**Panos Kalnis**
KAUST
panos.kalnis@kaust.edu.sa

## Abstract

Sparse tensors appear frequently in federated deep learning, either as a direct artifact of the deep neural network's gradients, or, as a result of an explicit sparsification process. Existing communication primitives are agnostic to the challenges of deep learning; consequently, they impose unnecessary communication overhead. This paper introduces DeepReduce, a versatile framework for the compressed communication of sparse tensors, tailored to federated deep learning. DeepReduce decomposes sparse tensors into two sets, values and indices, and allows both independent and combined compression of these sets. We support a variety of standard compressors, such as Deflate for values, and Run-Length Encoding for indices. We also propose two novel compression schemes that achieve superior results: curve-fitting based for values, and bloom-filter based for indices. DeepReduce is orthogonal to existing gradient sparsifiers and can be applied in conjunction with them, transparently to the end-user, to significantly lower the communication overhead. As a proof of concept, we implement our approach on TensorFlow and PyTorch. Our experiments with real models demonstrate that DeepReduce transmits 320% less data than existing sparsifiers, without affecting accuracy. Code is available at https://github.com/hangxu0304/DeepReduce.

## 1 Introduction

In federated learning [43, 47, 55, 72], the training is typically performed by a large number of resource-constrained client devices (e.g., smartphones), operating on their private data and computing a *local* model. Periodically, a subset of the devices is polled by a central server that retrieves their gradients, updates the *global* model and broadcasts it back to the clients. The most constrained resource in client devices is the network, either because the practically sustained bandwidth between remote clients and the server is low (typically, in the order of 25-50Mbps), or, the financial cost (e.g., data plans for 4G/5G mobile connections) is high. On the other hand, deep neural network model sizes have been steadily increasing at a much faster rate than the available bandwidth. Consequently, in federated learning, it is imperative to reduce the communicated data volume.

One key observation that can help with reducing this volume is that the data exchanged during the Deep Neural Network (*DNN*) training often correspond to *sparse* tensors, i.e., tensors with many zero-value elements. Sparse tensors may be: (*i*) direct artifacts of the training process; for instance, the gradients of the NCF [35] and DeepLight [18] models consist of roughly 40% and 99% zero elements, respectively; or (*ii*) explicitly generated by *sparsification* [6, 51, 69, 71, 76, 81],

35th Conference on Neural Information Processing Systems (NeurIPS 2021).

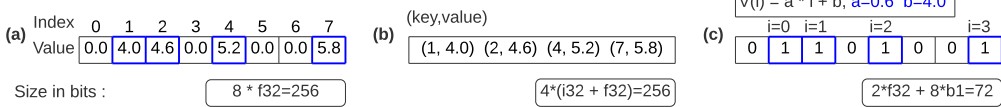

Figure 1: (a) Dense tensor format needs 256 bits; (b) sparse $\langle key, value \rangle$ form also needs 256 bits; (c) our method sends parameters, $(a, b)$ as well as an 8-bit string, i.e., 72 bits in total.

a commonly used lossy (e.g., Top-$r$ or Random-$r$) compression approach that selects only a few elements (with $r$ typically $< 1\%$).

Figure 1.a depicts an example tensor containing 8 real values, 4 of which are zero; its *dense* representation would require 256 bits. Typically, sparse tensors are represented as a set of $\langle key, value \rangle$ pairs (see Figure 1.b), where $key$ is the index; notice, however, that the $\langle key, value \rangle$ representation also requires 256 bits, negating the benefit of sparsity. We show an example of our improved approach in Figure 1.c. We consider the indices as an ordered list represented by a Boolean array of 8 bits, such that the $i^{\text{th}}$ bit is '1' if and only if the corresponding gradient element is non-zero. Moreover, we fit a function, $V(i) = a \cdot i + b$ to the gradient values, with parameters, $a = 0.6$ and $b = 4.0$. By transmitting only the bit string and parameters $(a, b)$, we can reconstruct the original tensor while requiring only 72 bits.

The above example demonstrates a significant margin for additional compression for sparse tensors. Recent works (e.g., SKCompress [40]) take advantage of these opportunities, but rely on a tightly coupled index and value compression algorithm that benefits only some scenarios (see Section 6). In practice, there exist complex trade-offs among data volume, model accuracy, and computational overhead, in conjunction with system aspects, such as the network bandwidth and the communication library (e.g., NCCL [2], or Gloo [29]). Given that no single solution fits all scenarios, practitioners need the flexibility to adjust how sparse tensors are compressed and transmitted for each particular DNN model and system configuration.

In this paper, we propose DeepReduce, a framework for transmitting sparse tensors via a wide-area network, tailored for large-scale federated DNN training. Our contributions include:

(*i*) The **DeepReduce framework**, described in Section 3, that decomposes the sparse tensor into two sets, indices and values, and allows for independent and combined compression. By decoupling indices from values, our framework enables synergistic combination of a variety of compressors in a way that benefits each particular DNN and training setup. DeepReduce resides between the machine learning framework (e.g., Tensorflow, PyTorch) and the communication library. It exposes an easy-to-use API that encapsulates a variety of existing methods, such as Run Length [83] and Huffman encoding [38] for index compression; as well as Deflate [20] and QSGD [7] for value compression. DeepReduce also provides an index reordering abstraction, which is useful for combining value with index compressors.

(*ii*) **Two novel compressors** for sparse tensors: a Bloom-filter based *index* compressor (Section 4) that reduces the size of keys by 50%, compared to the traditional $\langle key, value \rangle$ sparse representation; and a curve-fitting based *value* compressor (Section 5) that reduces the large values array to a small set of parameters. Both of our compressors do not affect the quality of the trained model.

(*iii*) An **evaluation** of DeepReduce on a variety of DNN models and applications (Section 6). We demonstrate the practical applicability of our framework by realistic *federated learning* deployments on geographically remote, large-scale cloud infrastructures, and show that DeepReduce compresses already-sparse data by up to 320%, without affecting the training quality.

## 2 Background

**Notations.** By $[d]$ we denote the set of $d$ natural numbers $\{1, 2, \cdots, d\}$. We denote the cardinality and complement of a set $X$, by $|X|$ and $X^c$, respectively; $x[i]$ is the $i^{\text{th}}$ component of vector $x$. By $x_S \in \mathbb{R}^d$ we denote an $|S|$-sparse vector, where $S \subseteq [d]$ is its support. $\|x\|$ and $\|x\|_\infty$ are the $\ell_2$ and $\ell_\infty$ norm of a vector $x$, respectively. Class $C^1$ consists of all differentiable functions with continuous derivative and $\text{Var}_{[a,b]}(f)$ denotes the variance of function $f \in C^1[a, b]$ over interval $[a, b]$.

**Federated Learning (*FL*)** trains models *collaboratively* on data stored in resource-constrained, heterogeneous and geographically remote client devices (e.g., smartphones). At each round, a fraction of the participating clients are selected. After synchronizing with the server-side *global* model, the selected clients perform *local* training[1] for some epochs over their private data. The server collects those updates, aggregates and applies them to the global model. Federated averaging (*FedAvg*) [55], is a popular instantiation of this process. Other variants exist, such as SCAFFOLD [44], FedNova [80], parallel SGD [85], local SGD [68], and FedProx [49]. Owing to insufficient communication bandwidth in FL, lossy compression is used to reduce the transmitted data volume; examples include FedBoost [31], FedCOM [30], and FedPAQ [60].

**Compressor** [69] is a random operator, $\mathcal{C}(\cdot) : \mathbb{R}^d \to \mathbb{R}^d$, that satisfies $\mathbb{E}_{\mathcal{C}} \|x - \mathcal{C}(x)\|^2 \leq \Omega \|x\|^2$, where $\Omega > 0$ is the compression factor and the expectation is taken over the randomness of $\mathcal{C}$. If $\Omega = 1 - \delta$ and $\delta \in (0, 1]$, $\mathcal{C}$ is a $\delta$-*compressor*, denoted by $\mathcal{C}_\delta$.

**Sparsification.** A rank-1 tensor that has mostly zero components is said to be sparse. Many modern DNNs are inherently sparse [18, 35, 42], and so are their gradients. Sparse gradients can also be generated by a compressor that sparsifies [51, 71, 76, 81] the gradient $g$ to generate $\tilde{g} \in \mathbb{R}^d$ via: $\tilde{g}[i] = \mathcal{C}(g[i]) = g[i]$, if $i \in S$, otherwise $\tilde{g}[i] = 0$. We assume that $S \subset [d]$ is the support set of $\tilde{g}$ such that $\tilde{g}$ is $r$-sparse if and only if $|S| = r$. Two commonly used sparsifiers, Random-$r$ [69] and Top-$r$ [6, 8], both are $\delta$-compressors. We refer to further details in Appendix A.

**Bloom filter** [12] is a probabilistic data structure that represents the elements of a set $S$. Initially, it is a bit string $\mathcal{B}$ of $m$ bits, all set to 0. To insert an element $y_j \in S$ in the Bloom filter, we apply $k$ independent hash functions $h_i$ on $y_j$. Each $h_i$ yields a bit location in $\mathcal{B}$, and changes that bit to 1. To query if $y_j \in S$, we apply all $k$ hash functions on it. If $h_i(y_j) = 0$ for any $i \in [k]$, then the element *surely* does not belong to $S$ (i.e., no false negatives). In contrast, if $h_i(y_j) = 1$ for all $i \in [k]$, then $y_j$ may or may not belong to $S$ (i.e., possible *false positives*). The false positive rate (*FPR*) [12] of $\mathcal{B}$ is: $\epsilon \approx (1 - e^{-k|S|/m})^k$; see Lemma 3 in Appendix A.2.

## 3 System Architecture

DeepReduce (see Figure 12 in the Appendix) resides between the machine learning framework (e.g., TensorFlow, PyTorch) and the communication library, and is optimized for federated DNN training. It offers a simple API whose functions can be overridden to implement, with minimal effort, a wide variety of index and value compression methods for sparse tensors. At the transmitting side, the input to DeepReduce is a sparse tensor directly from the ML framework, for the case of inherently sparse models; or, is generated by an explicit sparsification process. In our implementation, we employ the GRACE [84] library for the sparsification operation, since it includes many popular sparsifiers; other libraries can also be used.

Sparse tensors are typically represented as $\langle key, value \rangle$ tuples. DeepReduce decouples the keys from the values and constructs two separate data structures. Let $\tilde{g} \in \mathbb{R}^d$ be the sparse gradient, where $d$ is the number of model parameters and $\|\tilde{g}\|_0 = r$, is the number of nonzero gradient elements. Let $S$ be the set of $r$ indices corresponding to those elements. DeepReduce implements two equivalent representations of $S$: (*i*) an array of $r$ integers; and (*ii*) a bit string $B$ with $d$ bits, where $\forall i \in [1, d], B[i] = 1$ if and only if $\tilde{g}[i] \neq 0$. These two representations are useful for supporting a variety of index compressors; e.g., the bit string representation is used in [14]. The Index Compression module encapsulates the two representations and implements several algorithms for index compression. It supports both *lossy* compressors (e.g., our Bloom-filter based proposal), as well as *lossless* ones, such as the existing Run Length (RLE) [83] and Huffman [38, 27] encoders (see Appendix A.1); there is also an option to bypass index compression.

The Value Compression module receives the sparse gradient values and compresses them independently. Several compressors, such as Deflate [20], QSGD [7], and our own curve-fitting based methods, are implemented. Again, there is an option to bypass value compression. Some compressors (e.g., our own proposals), require reordering of the gradient elements, which is handled by the Index reorder module. DeepReduce then combines, in one container, the compressed index and value

---

[1]In contrast to FL, conventional data-parallel, distributed training [87] synchronizes the updates from *all* workers at *each* iteration (see Appendix A).

structures, the reordering information and any required metadata. Then, the container is passed to the communication library.

The receiving side mirrors the structure of the transmitter, but implements the inverse functions, that is, index and value decompression, and index reordering. The reconstructed sparse gradient is routed to GRACE for de-sparsification, or passed directly to the ML framework. It is worth mentioning that DeepReduce is general enough to represent popular existing methods that employ proprietary combined value and index compression. For example, SKCompress [40] can be implemented in DeepReduce as follows: SketchML [39] plus Huffman for values, no index reordering, and delta encoding plus Huffman for indices.

## 4  Bloom Filter for Indices

This section introduces our novel Bloom-filter based, lossy index compressor. Recall that $S$ is the set of $r$ indices (i.e., $|S| = r$) corresponding to the nonzero components of sparse gradient $\tilde{g}$. Let us insert each item of $S$ into a Bloom filter $\mathcal{B}$ of size $m$, using $k$ hash functions.

**Naïve Bloom filter.** Let $V$ be an array of size $r$ containing the elements of $\tilde{g}$ that are indexed by $S$; formally: $\forall i \in [1, r] : V(i) = \tilde{g}[S[i]]$. DeepReduce transmits $V$ and $\mathcal{B}$. The receiver initializes $ptr = 1$ and reconstructs the gradient as follows:

---
**for** $i = 1$ *to* $d$ **do**  /* all $d$ elements of gradient $\tilde{g} \in \mathbb{R}^d$ */
    **if** $i \in \mathcal{B}$ **then**  $\tilde{g}[i] = V[ptr]$; ptr++; **else** $\tilde{g}[i] = 0$

---

If $\mathcal{B}$ were *lossless*, the algorithm would have perfectly reconstructed the gradient. However, Bloom filters exhibit false positives (*FP*). Assume a single FP at $ptr = j$; then, for $j \leq ptr \leq r$, *every* $V[ptr]$ value will be assigned to the wrong gradient element. Therefore, FPs cause a disproportionately large error to the reconstructed gradient, significantly affecting the quality of the trained model.

**No-error approach: Policy P0.** To address this drawback of the naïve approach, we initialize a set $P = \varnothing$ and we modify the previous reconstruction algorithm as follows:

---
**for** $i = 1$ *to* $d$ **do**  /* all $d$ elements of gradient $\tilde{g} \in \mathbb{R}^d$ */
    **if** $i \in \mathcal{B}$ **then**  insert $i$ in $P$

---

$P$ contains the union of the true and false positive responses of $\mathcal{B}$. The transmitting worker can execute this algorithm and determine $P$ prior to any communication. With that information, it constructs array $V$ as follows: $\forall i \in [1, |P|] : V(i) = \tilde{g}[P[i]]$. Essentially, $V$ contains the gradient elements that correspond to both true and false positives. Therefore, the receiving worker can reconstruct the sparse gradient *perfectly*. The trade-off is increased data volume compared to Naïve, since the size of $V$ grows to $r \leq |P| \leq \lceil r + (\frac{1}{2})^{-\frac{\log(\epsilon)}{\log(2)}} (d - r) \rceil$; see Lemma 6 in Appendix B.2. We measure the compression error due to gradient, $\mathcal{C}_{P0,\delta}(g)$ resulted from policy $P0$ in Lemma 7 in Appendix B.2.

**Random approach: Policy P1.** To address the increased data volume issue of policy P0, this policy defines a new set of indices $\tilde{S} \subseteq P$, where $|\tilde{S}| = r$. $\tilde{S}$ is generated by randomly selecting $r$ elements from $P$. Consequently, array $V$ is constructed as $\forall i \in [1, r] : V(i) = g[\tilde{S}[i]]$. Since $\tilde{S} \neq S$ in general, we expected the error to be affected. Let $k_1 = |\tilde{S} \cap S|$ and assume that the input gradient is inherently sparse, i.e., sparsifier $\mathcal{C}_\delta$ is the identify function $I_d$. For the combined compressor $\mathcal{C}_{P1, I_d}$ with policy P1, we show in Appendix B.4 that $\mathbb{E}\|g - \mathcal{C}_{P1, I_d}(g)\|^2 = (1 - \frac{k_1}{r})\|g\|^2$. Specifically, policy P1 creates a lossy compressor with compression factor as good as Random-$k_1$. In practice, DeepReduce allows $\mathcal{C}_\delta$ to be any sparsifier (e.g., Top-$r$ [6, 8]). In this case, policy P1 is essentially equivalent to a combined sparsifier, similar to Elibol et al. [24] and Barnes et al. [9]. We give the total compression error due to the compressed gradient $\mathcal{C}_{\mathcal{P}_R, \delta}(g)$ and other detailed analysis and proofs of error bounds in Lemma 9 in Appendix B.4.

**Conflict sets: Policy P2.** P0 and P1 represent two extremes: P0 eliminates errors but sends more data than P1, whereas P1 sends fewer data but may introduce errors. Here, we propose policy P2, which transmits the same amount of data as P1, but is closer to P0 in terms of error. Similar to P1, this policy generates a set $\tilde{S} \subseteq P$, but makes better probabilistic choices. Intuitively, false positives are due to collisions in the Bloom filter, resulting in conflicts. P2 groups all items of $P$ into conflict sets. Two elements $x$ and $y$ belong to a conflict set $\mathcal{C}_j$ if $x, y \in P$ and $h_i(x) = h_{i'}(y) = j$ for $i, i' \in [k]$, where $j$ is the $j^{\text{th}}$ bit of $\mathcal{B}$ and $\bigcup_j \mathcal{C}_j = P$. Figure 2 shows an example, with 4 items and 3 hash

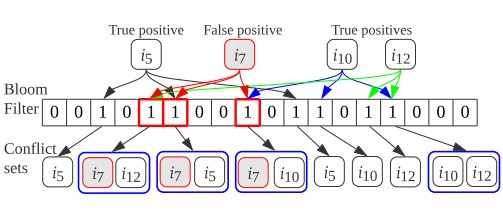

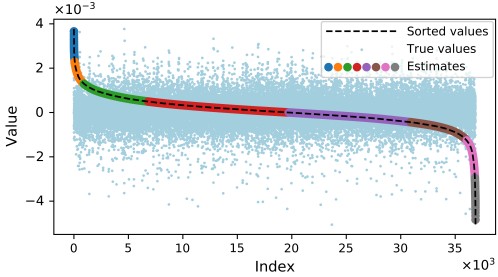

Figure 2: Policy P2, with 3 hash functions and 8 conflict sets.

Figure 3: Piece-wise (8 pieces) value fitting on a convolution layer gradient of ResNet20 (CIFAR-10).

functions; item $i_7$ is a false positive and there exist 8 conflict sets. If a set $C_j$ contains only one item, it is guaranteed to be a true positive, so it is added to $\tilde{S}$. Else, if $C_j$ contains many items, a random subset is inserted into $\tilde{S}$. Therefore, it is possible that some false positives are included in $\tilde{S}$, but the probability is smaller, compared to policy P1. We present the pseudo-code of policy P2 in Algorithm 1, Appendix B.5. For Hash functions used and an efficient GPU implementation on PyTorch, refer to Appendix E.

## 5 Curve Fitting for Values

Figure 3 shows, in light blue, the gradient values for one layer of ResNet20 on CIFAR-10. By sorting those values, we obtain a curve that can be approximated as a *smooth convex* curve. Taking advantage of this observation, we propose a novel curve fitting-based value compression approach that results into high compression ratios, even after considering the overhead of mapping the original to the sorted values. In contrast, modelling DNN gradients by using statistical distributions [26, 54, 66] is both restrictive and model specific.

Formally, we sort the nonzero values of the sparse gradient, $C_\delta(g_t)$ resulting from stochastic gradient, $g_t$ at each iteration $t$, in descending order and denote as $C_S(g_t) \in \mathbb{R}^d$. Let $C_S(g_t)$ follow the hierarchical model: $C_S(g_t) = \nabla f_t + \sigma \xi_t$, where $\xi_t \sim N(0, I_d)$, $\nabla f_t \in \mathbb{R}^d$ is the sorted oracle gradient and $\sigma \in \mathbb{R}^+$; the model is commonly used in signal processing for approximation problems [5, 41, 74]. With these assumptions, we show how to fit regression models on $\mathcal{D} := \{i, C_S(g_t)[i]\}_{i=1}^r$ and calculate the fitting error.

**Nonlinear approximation.** We employ a nonlinear approximation by using splines with free knots, which consists of splitting the sorted curve into segments and fitting each segment individually. Finding the locations of the knots is where the nonlinearity and difficulties are (cf. [16]). Although existing numerical algorithms can find the best splines with free knots, they are computationally intensive and unsuitable for our use at each iteration of DNN training. Because we aim at approximating a smooth convex curve, we use the following procedure by selecting the knot one by one. We explain how this is done for the positive sorted values; similar idea applies to the negative sorted values. Let the whole gradient be sorted in descending manner; set $[l]$, with $l \le d$ corresponds to indices of the sorted positive values. Let $y = mx + c$ be the line joining $(1, C_S(g)[1])$ and $(l, C_S(g)[l])$. Calculate $d_i := (y_i - C_S(g)[i])^2$, where $y_i = mi + c$. Choose the sorted gradient component that corresponds to $\max_{i \in [l]} d_i$ as the segmentation point (a knot). The process continues until the desired number of segments is reached. We stop segmentation if the number of points in a segment is less than $(n' + 1)$, where $n'$ is the degree of the polynomial used to fit on each segment. The following Proposition justifies our procedure above.

**Lemma 1.** *(Knot selection) If $f$ is a differentiable convex function on $[a, b]$, then the point $x^* \in (a, b)$ that gives the best approximation to $f$ using line segments from $(a, f(a))$ to $(x^*, f(x^*))$, and then from $(x^*, f(x^*))$ to $(b, f(b))$ is the same point $x^+$ that maximizes the difference between $f$ and the line segment connecting $(a, f(a))$ to $(b, f(b))$.*

**Polynomial regression.** Over each segment, we apply polynomial regression. Due to limited space, we focus on the piece-wise linear fit, and have the following result with *explicit constants*. For the results on piece-wise constant approximation, see Lemma 11 in Appendix C.

Table 1: Benchmarks and datasets; last column shows the best quality achieved by the no-compression baseline.

| Type | Model | Task | Dataset | Parameters | Optimizer | Platform | Metric | Baseline |
|---|---|---|---|---|---|---|---|---|
| CNN | ResNet-20 [34] | Image classif. | CIFAR-10 [48] | 269,722 | SGD-M [73] | TFlow | Top-1 Acc. | 90.94% |
| | DenseNet40-K12 [37] | Image classif. | CIFAR-10 [48] | 357,491 | SGD-M [73] | TFlow | Top-1 Acc. | 91.76% |
| | ResNet-50 [34] | Image classif. | ImageNet [17] | 25,557,032 | SGD-M [73] | TFlow | Top-1 Acc. | 73.78% |
| MLP | NCF [35] | Recommendation | Movielens-20M [56] | 31,832,577 | Adam [46] | PyTorch | Best Hit Rate | 94.97% |
| RNN | LSTM[59] | Next word pred. | Stack Overflow[67] | 4,053,428 | **FedAvg** [55] | PyTorch | Top-1 Acc. | 18.56% |

**Lemma 2.** *(Error of piece-wise linear fit) For $\mathcal{C}_S(g) \in C^1([1,d])$ with $\mathrm{Var}_{[1,d]}(\mathcal{C}'_S(g)) \leq M$, we have $\|s - \mathcal{C}_S(g)\|_\infty \leq \frac{2M}{p^2}$, for some $s \in \mathcal{S}^1_p$—the set of piece-wise linear splines with p knots.*

Note that, the least squares error is less than $\sqrt{d}\|s - \mathcal{C}_S(g)\|_\infty$, and can be controlled by using a large $p$ (see Remark 5 in the Appendix). We provide here a heuristic to calculate $p$ from Lemma 2. First, we calculate $M = |(\mathcal{C}_S(g)[1] - \mathcal{C}_S(g)[2]) - (\mathcal{C}_S(g)[d-1] - \mathcal{C}_S(g)[d])|$. By considering the error bound $\frac{2M}{p^2}$ as a function of $p$, we can find the closed-form solution for $p$ as $p = \lceil 2\sqrt{M} \rceil$.

**Nonlinear regression.** We can use nonlinear regression for value fitting, e.g., through a double exponential model, $y = ae^{bx} + ce^{dx}$, where $(a, b, c, d) \in \mathbb{R}^4$ are the parameters; refer to Section 6.

For the theoretical analysis of compression error from regression, see Appendix C.1. In Appendix D, we provide the compression error from joint value and index compression. Also, see Appendix D.1 for comments regarding the convergence of the approach. Finally, refer to Appendix E for an efficient GPU and CPU implementation of our polynomial regression on PyTorch and TensorFlow, and the combined index and value compression. For overall complexities of the compression methods we also refer to Appendix E.

## 6   Experimental Evaluation

**Implementation.** DeepReduce supports TensorFlow and Pytorch. We provide various versions of our index and value compressors, on CPUs and GPUs. We also instantiate our framework with combinations of existing methods, namely Huffman and RLE for index compression, as well as Deflate and QSGD [7] for value compression; see Appendix F. We denote implementations that use DeepReduce by $\mathrm{DR}^{val}_{idx}$, where $idx$, $val$ are the index and value compression methods, respectively.

**Testbed.** We set up a realistic federated learning testbed on Amazon AWS. The server is an EC2 instance located in Ohio, whereas the clients are 56 geographically remote instances spread across 7 regions throughout the globe (i.e., Tokyo, Central Canada, Northern California, Seoul, São Paulo, Paris and Oregon). Each instance is equipped with a 4-core Intel CPU @ 2.50GHz, 16GB RAM, and an NVIDIA Tesla T4 GPU with 16 GB on-board memory (see Appendix F.1 for details). We also run simulated deployments on a local cluster of 8 nodes, each with a 16-core Intel CPU @ 2.6GHz, 512GB RAM, one NVIDIA Tesla V100 GPU with 16 GB on-board memory and 100Gbps network.

**Benchmarks.** We employ the popular FedML [33] benchmark that uses an LSTM model [59] to perform next-word prediction in a federated learning setting, on the Stack Overflow [67] dataset with 135,818,730 training and 16,586,035 test examples; the dataset follows a real-life partitioning among 342,477 clients. We also use industry-standard benchmarks from TensorFlow [52, 75] and NVIDIA [57], on image classification and recomendation; refer to Table 1 for details.

### 6.1   Simulated deployment on a local testbed

First, we run a set of experiments on our local cluster to validate our index and value compressors.

**Bloom filter-based index compression.** Figure 4 depicts the convergence timeline for our three index compression polices for ResNet-20 on CIFAR-10; FPR is set to $10^{-3}$ (refer to Appendix F for the effect of varying FPR). We compare against the no-compression baseline, as well as the plain Top-$r$ sparsifier ($r = 1\%$). All our policies converge to the same top-1 accuracy as the no-compression baseline, but BF-P0 converges in fewer training epochs. It is worth noting that BF-P0 converges faster than the plain Top-$r$ sparsifier, despite transmitting 33% fewer data (refer to Figure 15c in the Appendix). Note that BF-naïve (Section 4) achieves much lower accuracy, justifying the need for our proposed policies. Similar results were observed for DenseNet40-K12; see Appendix F.3.

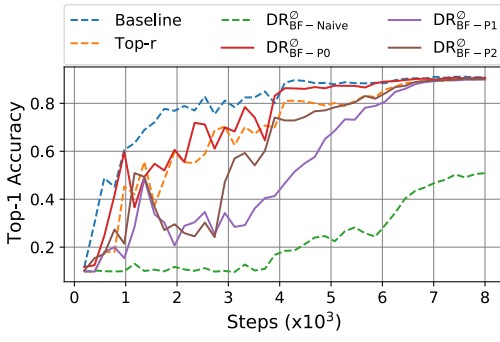
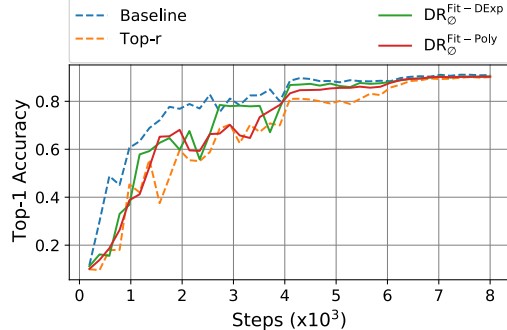

Figure 4: Convergence timeline of our bloom filter policies for ResNet-20 on CIFAR-10; FPR = 0.001, $r = 1\%$. $\mathrm{DR}_{\mathrm{BF-P0}}^{\varnothing}$ converges the fastest.

Figure 5: Convergence timeline of value compressors for ResNet-20 on CIFAR-10; $r = 1\%$. $\mathrm{DR}_{\varnothing}^{\mathrm{Fit-DExp}}$ is slightly better than $\mathrm{DR}_{\varnothing}^{\mathrm{Fit-Poly}}$.

**Curve fitting-based value compression.** Figure 5 shows the converge timeline for our two curve fitting-based value compressors, Fit-Poly and Fit-DExp, on ResNet-20 with CIFAR-10. The sparse input gradients were generated by Top-$r$ ($r = 1\%$). Fit-Poly uses polynomials with degree 5 (i.e., 6 coefficients). Fit-DExp requires 4 coefficients without segmentation. Both methods converge to the same accuracy as the no-compression baseline; also, both converge in fewer steps than plain Top-$r$. Fit-DExp sends fewer data: it compresses the output of Top-$r$ by roughly 50%, whereas Fit-Poly compresses it by around 40%. However, the computational overhead of compression for Fit-DExp is roughly $3.5\times$ more than Fit-Poly's; see details about the data volume and runtime in Figure 8.

**DeepReduce for an inherently sparse model.** DeepReduce is designed to work either in conjunction with a sparsifier (e.g., Top-$r$), or be applied directly on models such as the NCF [35] and DeepLight [18], that exhibit inherently sparse gradients. In this experiment, we train NCF on ML-20m, with $10^6$ local batch size. We test $\mathrm{DR}_{\mathrm{BF-P0}}^{\mathrm{QSGD}}$, which uses BF-P0 (FPR=0.6) for indices, but combines it with QSGD [7], an existing method for value compression. This demonstrates that DeepReduce is compatible with various existing compressors. We compare against SKCompress [40], an improved version of SketchML [39], optimized for sparse tensors. Table 6 in the Appendix shows the results. All methods achieve virtually the same best hit rate (i.e., the quality metric for NCF), and both $\mathrm{DR}_{\mathrm{BF-P0}}^{\mathrm{QSGD}}$ and SKCompress reduce the data volume by $5\times$ compared with Baseline. However, in practice $\mathrm{DR}_{\mathrm{BF-P0}}^{\mathrm{QSGD}}$ can be more easily implemented on GPUs; in Figure 8b we show that it is $380\times$ faster in terms of compression and decompression time. In Appendix F.3, we also compare DeepReduce against 3LC [50] and SketchML [39] on a larger benchmark, ResNet-50 on ImageNet.

## 6.2 Realistic Federated Learning deployment in the cloud

In this section, we use the FedML [33] benchmark to deploy a realistic federated learning system in the cloud. The real Stack Overflow [67] dataset is naturally partitioned among 342,477 clients. At each round, the server randomly selects 56 clients to communicate. Those clients are activated on physical EC2 instances in Amazon AWS, located at geographically remote data centers all over the world. Each client executes 1 local epoch; the learning rate is 0.3 and the batch size is 16. Sparsification is performed on tensors with more than 1 dimension, by bidirectional Top-$r$ ($r = 10\%$) compression with error feedback on the model updates. We execute 200 rounds and achieve 18.56% test accuracy, which is consistent with the FedML benchmark. We give pseudocode of FedAvg with DeepReduce in Algorithm 2 in Appendix F.3.

**Communication cost and computational overhead for compression.** Since the most constrained resource in federated learning is the network, our main target is to *minimize the amount of transferred data*. We measure separately the amount of transferred data from server-to-client (S2C) and client-to-server (C2S). Our baseline is the popular FedAvg [55] algorithm; we also compare against the plain Top-$r$ sparsifier ($r = 10\%$). Table 2 shows the results for three instantiations of DeepReduce. In all cases DeepReduce compresses significantly the already-sparse data. In particular, $\mathrm{DR}_{\mathrm{BF-P0}}^{\mathrm{QSGD}}$ transmits only 6.2% of the original data, which is $3.2\times$ less than Top-$r$, while the test accuracy

Table 2: Time breakdown and data volume of DeepReduce variants, Top-$r$, and Baseline (FedAvg [55]) in a FL setting. (CLI, SER, S2C and C2S stand for Client, Server, Server-to-Client, and Client-to-Server, respectively.)

| | Average Encoding/Decoding Time (s) | | | | Avg. Comm. Time (s) | | Avg. Data Volume (rel. to baseline) | | Test Accuracy |
|---|---|---|---|---|---|---|---|---|---|
| | $CLI_{decode}$ | $CLI_{encode}$ | $SER_{decode}$ | $SER_{encode}$ | S2C | C2S | S2C | C2S | |
| Baseline | 0 | 0 | 0 | 0 | 1.6014 | 1.6117 | 1.0 | 1.0 | 0.1856 |
| Top-$r$ (10%) | 0.0035 | 0.0266 | 0.0045 | 0.0299 | 0.7853 | 0.8165 | 0.2033 | 0.2033 | 0.1840 |
| $DR^{\varnothing}_{BF-P0}$ | 0.0181 | 0.0623 | 0.0161 | 0.0574 | 0.7763 | 0.7986 | **0.1425** | **0.1426** | 0.1841 |
| $DR^{Fit-Poly}_{\varnothing}$ | 0.0187 | 0.1178 | 0.0175 | 0.1024 | 0.6858 | 0.6876 | **0.1039** | **0.1039** | 0.1838 |
| $DR^{QSGD}_{BF-P0}$ | 0.0192 | 0.0754 | 0.0175 | 0.0691 | 0.6842 | 0.6864 | **0.0621** | **0.0621** | 0.1836 |

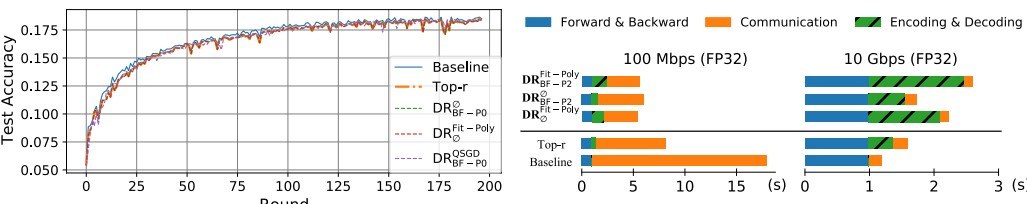

Figure 6: Convergence timeline of training an RNN to do next-word-prediction on Stack Overflow datasets. DeepReduce on Top-$r$ exhibits the same convergence rate as Top-$r$ in a FL setting.

Figure 7: Time breakdown in one iteration training of NCF on ml-20m. We show the speedup of training by DeepReduce on 4 nodes with different network bandwidth: 100Mbps vs. 10Gbps

remains virtually unaffected. The high quality of the resulting models is also confirmed in Figure 6, which shows the convergence timeline of the training. For completeness, Table 2 also reports the computational overhead for encoding and decoding, although this is not our primary concern. DeepReduce is 1.5-3.4× slower than Top-$r$, but in practice the overhead is within acceptable limits (i.e., less than 19msec). Despite the higher computational overhead, the average communication time is still up to 15% lower for DeepReduce compared to Top-$r$, and up to 2.3× lower compared to Baseline; the moderate gain is due to the high latency between geographically remote data centers in the Amazon AWS cloud. See test-accuracy vs. wall clock time, and server to client and client to server communication time (with error bars) in Figures 20 and 21, respectively, in Appendix F.

### 6.3 Practical applicability of DeepReduce

Any operation on the gradient imposes computational overheads that may exceed the benefits of the reduced data volume and affect the practical applicability, as discussed below.

**Suitability for federated learning.** We profile diverse deployments by training NCF on ML-20m and measuring the wall clock time of the various components. We employ gradient accumulation with 10 accumulations per iteration and $10^6$ local batch size. We use NCCL `Allreduce` for baseline communication, and NCCL `Allgather` for Top-$r$ ($r = 10\%$) and DeepReduce. We vary the network bandwidth from 100Mbps (typical for remote smartphones or IoT clients) up to 10Gbps (typical for local server clusters). Figure 7 shows the wall time, in three components: forward and back-propagation, encoding / decoding, and communication. Gradient compression is useful only when the ratio of communication over computation cost is high (i.e., lower bandwidth). This is consistent with the findings in [53, 58, 84] and reinforces our claim that DeepReduce is beneficial for federated learning deployments.

**Data volume and computational overhead.** In Figure 8a, we show the data volume (relative to the no-compression baseline) separately for values and indices, for various instantiations of DeepReduce. We test Resnet-20 on CIFAR-10 and generate sparse tensors by Top-$r$. We compare against SKCompress, which also operates on the sparse tensor. For fairness, parameters are selected such that all methods achieve similar accuracy. Although the ratio of index over value data volume differs for each combination of DeepReduce, all versions transmit fewer data than plain Top-$r$. Interestingly, SKCompress performs the best; however, this depends on the particular model.

Figure 8b shows, in logarithmic scale, the computational overhead of compression and decompression measured by the wall clock runtime. We implement all methods by utilizing the best available libraries either on CPUs or GPUs; we acknowledge there remains margin for improvement. Nonetheless, this

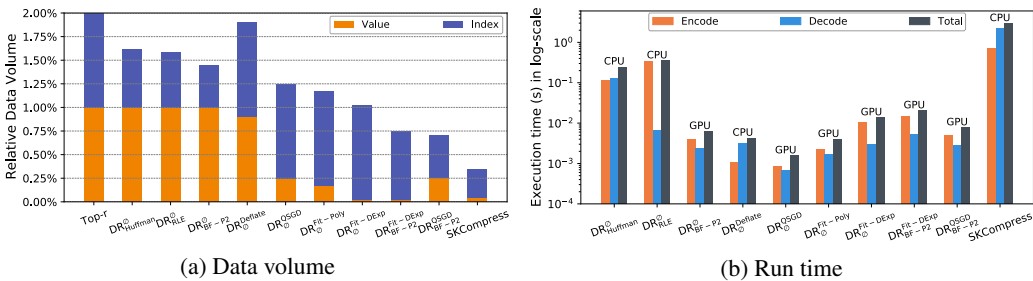

(a) Data volume        (b) Run time

Figure 8: Comparing various compression methods on the Top-$r$(1%) values of a convolution gradient in ResNet-20 (gradient size: 36,864). (a) Data volume (b) Encoding and decoding runtime

experiment demonstrates the significant variation in terms of overhead among methods; for instance, SKCompress is 3 orders of magnitude slower, compared to $DR_{\varnothing}^{QSGD}$.

**Discussion.** The previous experiment demonstrates that the practical benefit of any compressor, including DeepReduce, depends on multiple factors that affect the communication over computation ratio. Those include the communication library (e.g., NCCL, Gloo), the implementation details of each compressor, the computing hardware (e.g., faster GPUs), possible accelerators (e.g., NICs with FPGAs), and others. The advantage of DeepReduce lies in its versatility to intermix various index and value compressors, to match the requirements of each specific model and system configuration.

# 7 Related Work

**Gradient compression** is commonly employed to alleviate the network bottleneck in distributed training. Four main families of compression methods exist (refer to [84] for a survey): (*i*) Quantization [7, 11, 19, 45, 63, 82], where each tensor element is replaced by a lower precision one (e.g., float8 instead of float32), to achieve in practice compression ratios in the order of $4\times$ - $8\times$ [84]. (*ii*) *Sparsification* [6, 51, 69, 71, 76, 81], where only a few elements (e.g., Top-$r$ or Random-$r$) of the tensor are selected; it can achieve compression ratios of $100\times$ or more. (*iii*) Hybrid methods [10, 39, 50, 70], which combine quantization with sparsification to achieve a higher compression. (*iv*) Low-rank methods [15, 78, 79] that decompose the tensor into low-rank components.

**SketchML and SKCompress.** In SketchML [39], the nonzero gradient elements are quantized into buckets using a non-uniform quantile sketch. The number of buckets is further reduced via hash tables that resolve collisions by a Min-Max strategy. SKCompress [40] improves SketchML by additional Huffman coding on the bucket indices as well as the prefix of delta keys. Both of these methods can be viewed as special cases of DeepReduce.

**Hybrid compressors.** Qsparse local SGD [10] combines quantization with Top-$r$ or Random-$r$ sparsifiers. Strom et al. [70] and Dryden et al. [22] use a fixed and adaptive threshold, respectively, to sparsify. Elibol et al. [24] combine Top-$r$ with randomized unbiased coordinate descent. Barnes et al. in [9], perform a Top-$m$ selection of each local gradient and communicate $r < m$ randomly chosen components. Double quantization [86] is an asynchronous approach that integrates gradient sparsification with model parameter and gradient quantization. The output sparse gradient of hybrid methods can be the input to our framework; therefore, our work is orthogonal.

**Sparse tensor communication.** Communication libraries typically transmit sparse tensors via `Allgather` [1], because the more efficient `Allreduce` collective only supports dense tensors. In contrast, ScaleCom [13] tailors `Allreduce` to sparse data. OmniReduce [25] also implements sparse `Allreduce` that sends the non-zero blocks to the workers in an all-to-all manner. SparCML [61] adaptively switches between `Allreduce` and `Allgather` based on global gradient sparsity among the workers. SwitchML [62] is a hardware approach that aggregates the model updates in programmable network switches.

## 8 Conclusions

Sparse tensors are ubiquitous in federated DNN training. DeepReduce integrates seamlessly with popular machine learning frameworks and provides an easy-to-use API for the effortless implementation of a wide variety of sparse tensor communication methods. We instantiate DeepReduce both with existing index and value compressors, as well as with two novel methods: a Bloom filter-based index compressor and a curve fitting-based value compressor. We demonstrate its practical applicability by a realistic deployment in the cloud. DeepReduce is available as open-source and can be used to significantly lower the communication overhead in large-scale federated learning deployments.

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
