# A Background

This section complements Section 2 in the main paper by incorporating extra details (Sections A.1 and A.2) and technicalities (Section A.3 and A.4). These are instrumental for the rest of the paper.

**Compressed distributed training via SGD.** One can consider the traditional data-center DNN training as a special form of FL training, but without privacy. That is, the dataset is partitioned into all participating compute nodes. Moreover, in contrast to a fraction of participating clients in FL, all nodes update the model parameter, $x$. Additionally, the synchronization happens at each iteration, instead of a few local epochs at the nodes. Formally, during back-propagation with $n$ workers (i.e., compute nodes), to update the model parameter $x$, each worker $i$, at each iteration, calculates a stochastic gradient $g_t^i$ by processing an independent batch of data, $D_i$ with $\bigcup_i D_i = D$, the global dataset. Often, for efficient communication, the gradient is compressed to $\tilde{g}_t^i$ and is communicated to all workers, either through a parameter server [58], or through a peer-to-peer collective, like `Allreduce` [64]. The aggregated gradient, $\tilde{g}_t = \frac{1}{n} \sum_{i=1}^n \tilde{g}_t^i$ is then transmitted to all workers, who update the parameters of their local model via: $x_{t+1} = x_t - \eta_t \tilde{g}_t$, where $\eta_t > 0$ is the learning rate; the process repeats until convergence. Figure 9 shows an example of distributed training for DNNs with compressed communication.

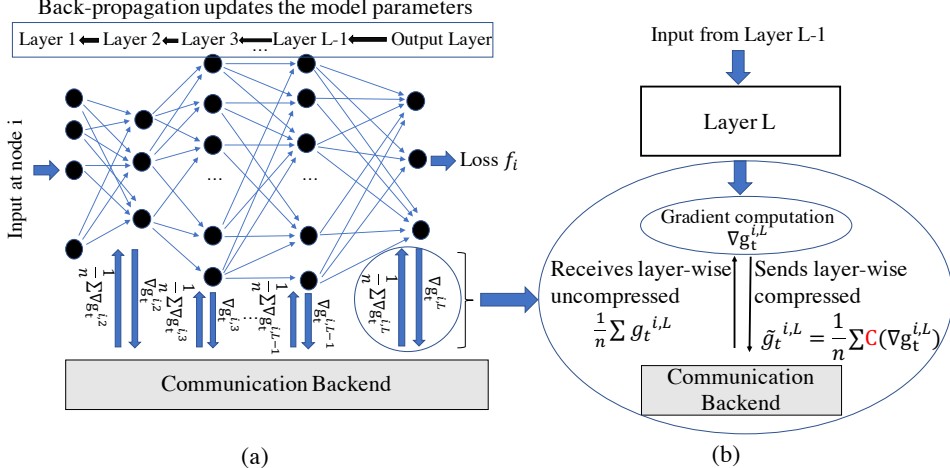

Figure 9: Distributed training from the perspective of $i^{\text{th}}$ computing node.

**Random-$r$, Top-$r$ sparsifiers, and $\delta$-compressors.** Based on the selection criteria of the elements in $S$, some of the most commonly used sparsifiers such as, Random-$r$ [69] and Top-$r$ [6, 8] are defined. For Random-$r$, the elements of $S$ are randomly selected out of $[d]$, whereas, for Top-$r$, the elements of $S$ correspond to the indices of the $r$ highest magnitude elements in $g$. Moreover, sparsifiers that follow (6) with $\Omega = 1 - \delta$, and $\delta \in (0, 1]$, are known as $\delta$-*compressors* and denoted by $\mathcal{C}_\delta$. That is,

$$\mathbb{E}\|g - \mathcal{C}_\delta(g)\|^2 \leq (1 - \delta)\|g\|^2. \tag{1}$$

*Remark* 1. Both Top-$r$ and Random-$r$ are $\delta$-*compressors* with $\delta = \frac{r}{d}$, and $\mathbb{E}\|g - \text{Top}r(g)\|^2 \leq \mathbb{E}\|g - \text{Random}r(g)\|^2 = (1 - r/d)\|g\|^2$, for all $g \in \mathbb{R}^d$.

## A.1 Lossless encoding strategies

In this section, we explain two *lossless* strategies that can be used in the DeepReduce framework. Discussion pertaining to their implementation is given in Section F.2.

**Run Length Encoding (RLE)** [83] is a *lossless* compressor in which consecutive occurrences of symbols are encoded as $\langle frequency, symbol \rangle$ tuples. For example, string "$aaaabaa$" is encoded as: $(4, "a"), (1, "b"), (2, "a")$. RLE is used to compress large sequences of repetitive data. In this work, we employ bit-level RLE, where symbols are 0 or 1, for index compression.

**Huffman encoding** [38] is a *lossless* scheme that assigns the optimal average decode-length pre-fix codes, using a greedy algorithm to construct a Huffman code tree. Higher frequency symbols are encoding with fewer bits. For instance, string "*aaaabaacaabaa*" generates mapping ("a", "b", "c") → (0, 10, 11) resulting to the following encoding: 0000**100011001000. Huffman encoding has been used to compress DNN weights [28, 32], as well as sparse gradient indices (e.g., SKCompress [40]).

## A.2 Further details on classic Bloom filter

The following Lemma characterizes the probability of the false-positive rates in a Bloom filter and Figure 10 is an example of a Bloom filter.

**Lemma 3.** *[12] Let $k$ denote the number of independent hash functions, $m$ the dimension of the bit-string, and $r$ the cardinality of the index set, $S$. Then the probability of the false-positive rate (FPR) is $\epsilon \approx (1 - e^{-kr/m})^k$.*

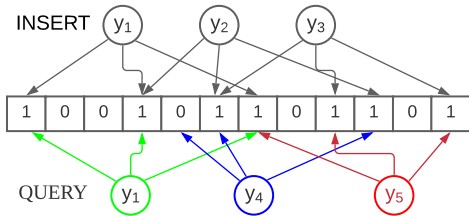

Figure 10: The figure illustrates an example Bloom filter $\mathcal{B}$ with $m = 12$ bits and $k = 3$ hash functions, representing a set, $S = \{y_1, y_2, y_3\}$. During querying, $y_1$ and $y_4$ are correctly identified as belonging (i.e., true positive) and not belonging (i.e., true negative) to $\mathcal{B}$, respectively. In contrast, $y_5$ is wrongly identified (i.e., false positive) as belonging to $\mathcal{B}$.

*Remark* 2. Given $\epsilon$ and $r$, the optimal $m = -\frac{r \log \epsilon}{(\log 2)^2}$ and $k = -\frac{\log \epsilon}{\log 2}$. Given $m$ and $r$, the number of hash functions that minimizes the probability of false positives is $k = \frac{m}{r} \log 2$. This $k$ results in the probability of false positive, $\epsilon$ as $\log \epsilon = -\frac{m}{r}(\log 2)^2$. In practice, we need to calculate the bits in the filter by using the relation $m = \frac{-r \log \epsilon}{(\log 2)^2}$ and the number of hash functions by $k = \frac{-\log \epsilon}{\log 2}$.

## A.3 Inequalities used in this paper

1. If $a, b \in \mathbb{R}^d$ then the Peter-Paul inequality is: There exists a $\xi > 0$ such that
$$\|a + b\|^2 \leq (1 + \xi)\|a\|^2 + (1 + \frac{1}{\xi})\|b\|^2. \tag{2}$$

We generally use a relaxed version of the above inequality as follows:
$$\|a + b\|^2 \leq 2\|a\|^2 + 2\|b\|^2. \tag{3}$$

2. If $a, b \in \mathbb{R}^d$ then we have
$$2\langle a, b \rangle \leq 2\|a\|^2 + \frac{1}{2}\|b\|^2. \tag{4}$$

3. For $x_i \in \mathbb{R}^d$ we have:
$$\|\sum_{i=1}^{n} x_i\|^2 \leq n \sum_{i=1}^{n} \|x_i\|^2. \tag{5}$$

4. If the operator $\mathcal{C} : \mathbb{R}^d \to \mathbb{R}^d$ is a *compressor* then there exists $\Omega > 0$ such that
$$\mathbb{E}\|g - \mathcal{C}(g)\|^2 \leq \Omega\|g\|^2. \tag{6}$$

5. If $X$ is a random variable then:
$$\mathbb{E}\|X\|^2 = \|\mathbb{E}[X]\|^2 + \underbrace{\mathbb{E}[\|X - E[X]\|^2]}_{\text{Var}(X)}. \tag{7}$$

## A.4 Preliminary results

The next two Lemmas are instrumental in proving other compression related results.

**Lemma 4.** *Let $x \in \mathbb{R}^d$ and $x_S$ be a vector that has the components of $x$ arranged in ascending/descending order of magnitude. If $0 \leq \theta < \pi/2$ be the angle between $x$ and $x_S$, then $\|x - x_S\|^2 = 2(1 - \cos\theta)\|x\|^2$.*

*Proof.* We have

$$\|x - x_S\|^2 = \|x\|^2 + \|x_S\|^2 - 2\langle x, x_S \rangle \overset{\|x\|=\|x_S\|}{=} 2\|x\|^2 - 2\|x\|^2 \cos\theta = 2(1 - \cos\theta)\|x\|^2.$$

Hence the result. □

**Lemma 5.** *Let $\mathcal{C}(\cdot) : \mathbb{R}^d \to \mathbb{R}^d$ be a $\delta$-compressor.*

*(i) If $\mathcal{C}_\delta(g)$ is unbiased then $\mathbb{E}\|\mathcal{C}_\delta(g)\|^2 \leq (2 - \delta)\|g\|^2$.*

*(ii) If $\mathcal{C}_\delta(g)$ is biased then $\mathbb{E}\|\mathcal{C}_\delta(g)\|^2 \leq 2(2 - \delta)\|g\|^2$.*

*Proof.* (*i*) Recall from (1), for $\delta$-compressors, we have $\mathbb{E}\|g - \mathcal{C}_\delta(g)\|^2 \leq (1 - \delta)\|g\|^2$. Since $\mathbb{E}(\mathcal{C}_\delta(g)) = g$, from (7) we have,

$$\mathbb{E}\|\mathcal{C}_\delta(g)\|^2 \overset{\text{By (7)}}{=} \mathbb{E}\|g - \mathcal{C}_\delta(g)\|^2 + \|g\|^2 \overset{\text{By (1)}}{\leq} (1 - \delta)\|g\|^2 + \|g\|^2 = (2 - \delta)\|g\|^2.$$

(*ii*) On the other hand, for biased compressors, by (3) we have,

$$\mathbb{E}\|\mathcal{C}_\delta(g)\|^2 = \mathbb{E}\|g - g + \mathcal{C}_\delta(g)\|^2 \overset{\text{By (3)}}{\leq} 2\mathbb{E}\|g - \mathcal{C}_\delta(g)\|^2 + 2\|g\|^2 \overset{\text{By (1)}}{\leq} 2(1-\delta)\|g\|^2 + 2\|g\|^2 = 2(2-\delta)\|g\|^2.$$
□

# B Bloom filter based index compression

In this section, we discuss in details different Bloom filter policies.

**Overview.** This Section serves as an addendum to Section 4 in the main paper and incorporates detailed discussions, examples, pseudocode, theoretical results, and their proofs. We start with an example in B.1 to illustrate the Naïve compression. Section B.2 provides proofs the Lemmas discussed in the main paper related to policy $P0$. In Section B.3, we discussed a new policy, called deterministic policy that sets the stage for more complex policies, random approach, Policy $P1$ (Section B.4) and conflict sets policy, Policy $P2$ (Section B.5).

## B.1 Naïve compression

In this scope we explain Naïve compression by a simple illustration, see Figure 11. We have a dense gradient represented as a sparse tensor in a key-value format. Notice that, the values in the sparse representation are sorted by their indices in an increasing order. The set, $S$ of keys is represented as a bloom filter. To communicate the sparse tensor we send both the values and the bloom filter. During the phase of decompression, we try to reconstruct $S$ by following the process we described. However, in this case, we manage to retrieve only 4 out of the 5 elements of $S$. Index 4 does not belong to $S$ and corresponds to a FP response. The mapping, $\mathcal{M}$ scans the communicated values in the order they arrive and assigns each one of them to the next larger index from the set of decoded indices. Notice how the selection of one wrong index affects the decompression by causing re-arrangements or shifts of the reconstructed gradient components with respect to their true positions.

## B.2 Policy P0

We provide the theoretical results involving policy $P0$ stated in the main paper.

**Lemma 6.** *The cardinality of the set $P$ is at most $\lceil r + (\frac{1}{2})^{-\frac{\log(\epsilon)}{\log(2)}}(d - r) \rceil$ and approaches to $r$ as $\epsilon \to 0$.*

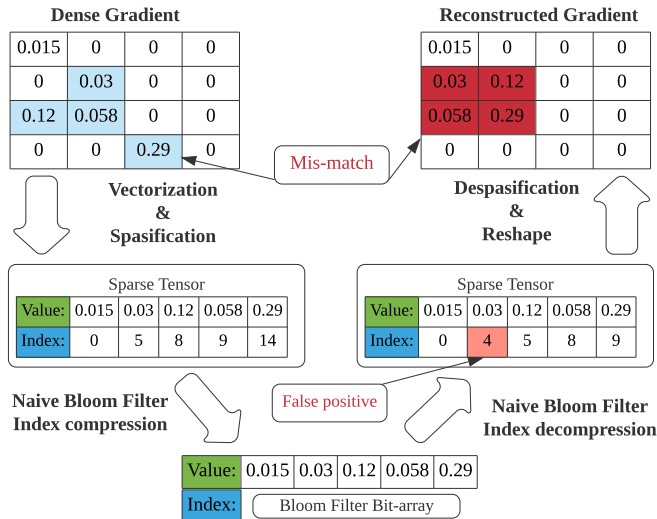

Figure 11: Naïve Bloom filter example that demonstrates how the FP elements in the Bloom filter, can cause re-arrangements or shifts of the reconstructed gradient components with respect to their true positions.

*Proof.* For given $\epsilon, r, d$ the cardinality of the set of positives $P$ follows

$$r \leq |P| \leq \lceil r + \epsilon(d - r) \rceil \overset{\text{By Lemma 3}}{\approx} \lceil r + (1 - e^{-kr/m})^k (d - r) \rceil.$$

Given $\epsilon$ and $r$, the optimal $m = -\frac{r \log \epsilon}{(\log 2)^2}$ and $k = -\frac{\log \epsilon}{\log 2}$. Therefore, plugging them in the above expression, we get

$$r \leq |P| \leq \lceil r + \left(\frac{1}{2}\right)^{-\frac{\log(\epsilon)}{\log(2)}} (d - r) \rceil,$$

which after taking limit $\epsilon \to 0$, gives the desired result. $\qquad\square$

The next Lemma measures compression error due to compressed gradient, $\mathcal{C}_{\mathcal{P}_0, \delta}(g)$.

**Lemma 7.** *(i) For a general $\delta$-compressor, $\mathcal{C}_\delta$, there exists a $\beta \in [0, 1)$, $\beta \geq \delta$ such that the compression error due to a compressed gradient, $\mathcal{C}_{\mathcal{P}_0, \delta}(g)$ resulted from $\mathcal{P}_0$ is $\mathbb{E}\|g - \mathcal{C}_{\mathcal{P}_0, \delta}(g)\|^2 \leq (1 - \beta)\|g\|^2$. (ii) For inherently sparse gradient, g, with $\mathcal{C}_\delta = I_d$, we have $\beta = \delta = 1$.*

*Proof.* (i) Consider a $\delta$ sparsifier, $\mathcal{C}_\delta$ such that $\|\mathcal{C}_\delta(g)\|_0 = r$. If $\mathcal{P}_0$ is used then, by Lemma 6, $|P| = r + (\frac{1}{2})^{-\frac{\log \epsilon}{\log 2}}(d - r) \geq r$ for $\epsilon \geq 0$, ($|P| = r$ for $\epsilon = 0$). Therefore, for $\mathcal{C}_{\mathcal{P}_0, \delta}$, we have $\beta = |P|/d > \delta$ resulting $\mathbb{E}\|g - \mathcal{C}_{\mathcal{P}_0, \delta}(g)\|^2 \leq (1 - \beta)\|g\|^2$.

*(ii)* For inherently sparse gradient, $g$, with $\mathcal{C}_\delta = I_d$, we have $\delta = 1$, and $\|\mathcal{C}_{I_d}(g)\|_0 = r$. Therefore, for policy $\mathcal{P}_0$, we have $\|\mathcal{C}_{\mathcal{P}_0, I_d}(g)\|_0 = r$ resulting $\beta = \delta = 1$.

$\qquad\square$

*Remark* 3. We consider two extreme cases of $\delta$ sparsifier. For Random-$r$, $\delta = r/d \leq 1$. If $\mathcal{P}_0$ is used then $|P| = r + (\frac{1}{2})^{-\frac{\log \epsilon}{\log 2}}(d - r) > r$ for $\epsilon > 0$. In this case, for $\mathcal{C}_{\mathcal{P}_0, \delta}$ we have $\beta = |P|/d > \delta$.

In another extreme case, for $\mathcal{C}_\delta$ to be Top-$r$, by Remark 1, $\delta \leq r/d \leq 1$. For $g \in \mathbb{R}^d$, similar argument as above gives us:

$$\mathbb{E}\|g - \mathcal{C}_{\mathcal{P}_0, \delta}\|^2 < \mathbb{E}\|g - \text{Top}r(g)\|^2 \leq \mathbb{E}\|g - \text{random}r(g)\|^2 \leq (1 - r/d)\|g\|^2.$$

Lemma 6 show that for small $\epsilon$, no policy sends negligible amount of extra data compared to the other policies. The GRACE [84] sparsification library allows to use the original dense gradient $g$, instead of $\tilde{g}$, to populate $V$. Consequently, all elements corresponding to false positives (i.e., set

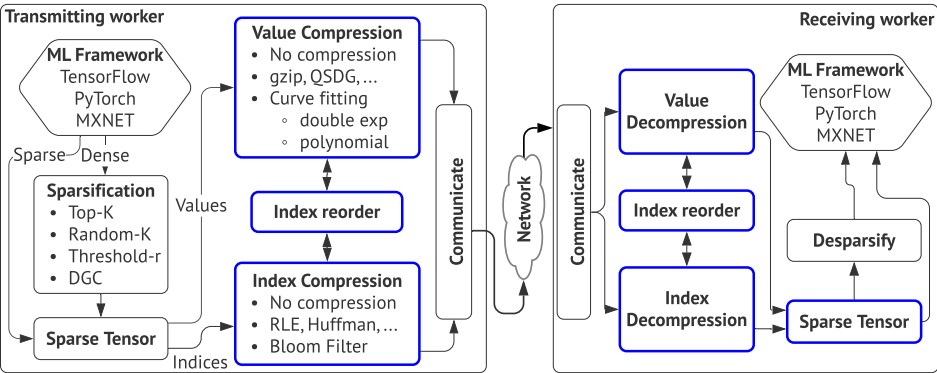

Figure 12: DeepReduce system architecture (highlighted in blue). DeepReduce resides between the machine learning framework (e.g., TensorFlow, PyTorch) and the communication library, and is optimized for federated DNN training. It offers a simple API whose functions can be overridden to implement, with minimal effort, a wide variety of index and value compression methods for sparse tensors. At the transmitting worker side, the input to DeepReduce is a sparse tensor, which is fed directly from the ML framework, for the case of inherently sparse models; or, is generated by an explicit sparsification process. Sparse tensors are typically represented as $\langle key, value \rangle$ tuples. DeepReduce decouples the keys from the values and constructs two separate data structures. Let $\tilde{g} \in \mathbb{R}^d$ be the sparse gradient, where $d$ is the number of model parameters and $\|\tilde{g}\|_0 = r$, is the number of nonzero gradient elements. Let $S$ be the set of $r$ indices corresponding to those elements. DeepReduce implements two equivalent representations of $S$: (*i*) an array of $r$ integers; and (*ii*) a bit string $B$ with $d$ bits, where $\forall i \in [1, d], B[i] = 1$ if and only if $\tilde{g}[i] \neq 0$. These two representations are useful for supporting a variety of index compressors. The Index Compression module encapsulates the two representations and implements several algorithms for index compression. It supports both *lossy* compressors (e.g., our Bloom filter-based proposal), as well as *lossless* ones, such as the existing Run Length (RLE) [83] and Huffman [38, 27] encoders; there is also an option to bypass index compression. The Value Compression module receives the sparse gradient values and compresses them independently. Several compressors, such as Deflate [20] and QSGD [7], are implemented, in addition to our own curve fitting-based method. Again, there is an option to bypass value compression. Some value compressors (e.g., our own proposals), require reordering of the gradient elements, which is handled by the Index reorder module. DeepReduce then combines in one container the compressed index and value structures, the reordering information and any required metadata; the container is passed to the communication library. The receiving worker, at the right of the figure, mirrors the structure of the transmitter, but implements the reverse functions, that is, index and value decompression, and index reordering. The reconstructed sparse gradient is routed for de-sparsification, or passed directly to the ML framework. DeepReduce is general enough to represent popular existing methods that employ proprietary combined value and index compression. E.g., SKCompress [40] can be implemented in DeepReduce as follows: SketchML [39] plus Huffman for values, no index reordering, and delta encoding plus Huffman for indices.

$P - S$) receive the original, instead of zero values. Lemma 7 (*i*) shows that for sparsified vectors, no policy achieves a better compression factor than the original sparsifier, $\mathcal{C}_\delta$. However, for inherently sparse tensors, Lemma 7 (*ii*) shows that no policy is *lossless* and is the best choice.

### B.3 Deterministic policy

This policy deterministically selects a subset of $r$ elements from $P$ and is denoted by $\mathcal{P}_D$. One can select the first $r$, the middle $r$, or the last $r$ elements from $P$, and based on this denote them as, leftmost-$r$, middle-$r$, and rightmost-$r$ policy, respectively. For implementation, the set $\tilde{S}$ can be created while iterating and posing queries on the universe $U$ and once it has $r$ elements, the querying is stopped. Let $\mathcal{C}_\delta$ be a general $\delta$-compressor that selects $r$ gradient components. However, with a policy, $\mathcal{P}_D$, not all the $r$ selected indices are due to $\mathcal{C}_\delta$. Let $I_1$ denote the set of indices that are selected via policy $\mathcal{P}_D$ originally resulted from $\mathcal{C}_\delta$ sparsifier and let $I_2$ denote the set of the rest of the $(r - |I_1|)$ indices. Therefore, $\tilde{S} = I_1 \bigcup I_2$ and let $\mathcal{C}_{\mathcal{P}_D, \delta}(g)$ be the compressed gradient whose indices are drawn via policy $\mathcal{P}_D$ and has support $\tilde{S}$. The following lemma quantifies the compression error.

**Deterministic policy error.** Lemma 8 gives the compression error bound for Deterministic policies.

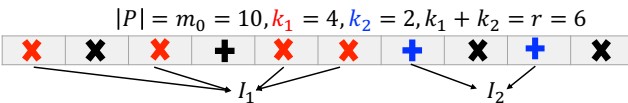

Figure 13: Random policy, policy $P0$ example with size of $|P| = m_0 = 10$, ✖ and ✚ denote TP and FP, respectively. Policy $P1$ selects set $I_1$ (in ✖), with $k_1 = 4$ elements and $I_2$ (in ✚), with $k_2 = 2$ elements, and $r = 6$.

**Lemma 8.** (*i*) *For all deterministic policies, $\mathcal{P}_D$ and for an inherently sparse gradient, $g$, the compression error due to a compressed gradient, $\mathcal{C}_{\mathcal{P}_D, I_d}(g)$ is given by $\mathbb{E}_{\tilde{S}}\|g - \mathcal{C}_{\mathcal{P}_D, I_d}(g)\|^2 = (1 - \frac{|I_1|}{r})\|g\|^2$.*

(*ii*) *For all deterministic policies, $\mathcal{P}_D$ and a general $\mathcal{C}_\delta$, the compression error due to a compressed gradient, $\mathcal{C}_{\mathcal{P}_D, \delta}(g)$ is given by $\mathbb{E}_{\tilde{S}}\|g - \mathcal{C}_{\mathcal{P}_D, \delta}(g)\|^2 = (1 - \frac{r}{d})\|g\|^2$.*

The proof of Lemma 8 follows from the standard procedure of taking expectation with respect to all possible $|I_1|$ cardinality subsets formed from the set $P$ by using policy, $\mathcal{P}_D$ and follows the same structure as the proof of Lemma 9 (*i*) and (*ii*). We omit the proof.

Lemma 8 (*i*) shows that by adopting a deterministic policy, $\mathcal{P}_D$ on the support of a general $\delta$-compressor, the compression error in expectation is as good as using a Random-$r$ sparsifier on the original gradient. Moreover, Lemma 8 (*ii*) shows that by adopting a deterministic policy, $\mathcal{P}_D$ on the support of a inherently sparse vector, the compression error in expectation is as good as using a Random-$|I_1|$ sparsifier on the original gradient. Therefore, it creates a *lossy* compression whose compression factor is as good as a Random-$|I_1|$ compressor on $r$ elements.

### B.4   Random Policy: Policy P1

First, we define the random policy, Policy $P1$ in detail, and then inspect the error incurred due to $P1$.

**Random approach: Policy P1.** A deterministic policy, $\mathcal{P}_D$ may be prone to bias, based on how the gradient components are distributed and the sparsifier used. E.g. If the support set of the sparsifier, $\mathcal{C}_\delta$ is concentrated at the beginning of $P$, then except leftmost-$r$, other deterministic policies such as, middle-$r$ and rightmost-$r$ incur more bias as they select more elements in $I_2$ than $I_1$. Similarly, if the support set of the sparsifier, $\mathcal{C}_\delta$, concentrated at the center, then only middle-$r$ policy is expected to incur the least bias compare to the others as it selects more elements in $I_1$ than in $I_2$. Without knowing the distribution of the gradient components, it is hard to invoke a deterministic policy. A random policy, $\mathcal{P}_R$ forms $\tilde{S}$ by picking $r$ indices randomly from $P$. Without loss of generality, consider the only source of randomness here is due to the random selection of $r$ indices and is unaffected by other source of randomness—randomness in the i.i.d. data, independence of the hash functions, etc.

Let $|P| = m_0$. Let policy $\mathcal{P}_R$ chooses a set $I_1$ of $k_1$ elements from the support set of a $\delta$-compressor $\mathcal{C}_\delta$ without replacement. Let the rest $k_2$ elements belong to the set $I_2$ such that, $k_1 + k_2 = r$. We illustrate this in Figure 13. By this, we incur two types of errors with respect to the vector $g_P$. The first error, $E_1$, is due to the compressed gradient $g_{I_1}$. The second error, $E_2$, is due to the compressed gradient $g_{I_2}$. We have, $\mathcal{C}_{\mathcal{P}_R, \delta}(g) := g_{I_1} \bigoplus g_{I_2}$. In the following Lemma, we measure the compression error due to compressed gradient $\mathcal{C}_{\mathcal{P}_R, \delta}(g)$ with respect to $g_P$.

**Lemma 9.** *With the notations mentioned above, we have the following measures of the compression error:*
(i) $E_1 = \mathbb{E}_{\mathcal{P}_R}\|g_P - g_{I_1}\|^2 = (1 - \frac{k_1}{r})\|g_P\|^2$.
(ii) $E_2 = \mathbb{E}_{\mathcal{P}_R}\|g_P - g_{I_2}\|^2 = (1 - \frac{k_2}{m_0 - r})\|g_P\|^2$.
(iii) *Denote $E := \mathbb{E}_{\mathcal{P}_R}\|g_P - \mathcal{C}_{\mathcal{P}_R, \delta}(g)\|^2$ be the total compression error due to the compressed gradient $\mathcal{C}_{\mathcal{P}_R, \delta}(g)$ with respect to $g_P$. Then, $E \leq E_1 + E_2$.*

*Proof.* Let $\Omega_{k_1}$ and $\Omega_{k_2}$ denote the set of all $k_1$ and $k_2$ elements subsets of the sets having cardinality $r$ and $m_0 - r$, respectively. The first error, $E_1$, is due to the compressed gradient $g_{I_1}$ whose support belongs to $\Omega_{k_1}$. The second error, $E_2$, is due to the compressed gradient $g_{I_2}$ whose support belongs to $\Omega_{k_2}$. With the notations mentioned above, we have

*(i)*

$$E_1 = \mathbb{E}_{\mathcal{P}_R}\|g_P - g_{I_1}\|^2 = \frac{1}{|\Omega_{k_1}|}\sum_{I_1 \in \Omega_{k_1}}\sum_{i=1}^{m_0}g_i^2 \mathbb{I}\{i \notin I_1\} = \|g_P\|^2\left(\frac{1}{\binom{r}{k_1}}\frac{r-k_1}{r}\binom{r}{k_1}\right) = (1-\frac{k_1}{r})\|g_P\|^2.$$

*(ii)* Similarly, we have $E_2 = \mathbb{E}_{\mathcal{P}_R}\|g_P - g_{I_2}\|^2 = \frac{1}{|\Omega_{k_2}|}\sum_{I_2 \in \Omega_{k_2}}\sum_{i=1}^{m_0}g_i^2\mathbb{I}\{i \notin I_2\} = $
$\|g_P\|^2\left(\frac{1}{\binom{m_0-r}{k_2}}\frac{m_0-r-k_2}{m_0-r}\binom{m_0-r}{k_2}\right) = (1 - \frac{k_2}{m-r})\|g_P\|^2.$

*(iii)* Denote $E := \mathbb{E}_{\mathcal{P}_R}\|g_P - \mathcal{C}_{\mathcal{P}_R,\delta}(g)\|^2$ be the total compression error due to the compressed gradient $\mathcal{C}_{\mathcal{P}_R,\delta}(g)$ with respect to $g_P$. Then, by using the linearity of expectation, we have

$$\begin{aligned}
E \quad &= \quad &&\mathbb{E}_{\mathcal{P}_R}\|g_P - \mathcal{C}_{\mathcal{P}_R,\delta}(g)\|^2 \\
&= \quad &&\mathbb{E}_{\mathcal{P}_R}\|g_P - g_{I_1}\bigoplus g_{I_2}\|^2 \\
&\overset{\langle g_P, g_{I_1}\bigoplus g_{I_2}\rangle = \sum_{i \in I_1 \bigcup I_2}g_i^2}{=} \quad &&\mathbb{E}_{\mathcal{P}_R}\|g_P\|^2 + \mathbb{E}_{\mathcal{P}_R}\|g_{I_1}\bigoplus g_{I_2}\|^2 - 2\mathbb{E}_{\mathcal{P}_R}\left(\sum_{i\in I_1\bigcup I_2}g_i^2\right) \\
&\overset{\langle g_{I_1}, g_{I_2}\rangle = 0}{=} \quad &&\mathbb{E}_{\mathcal{P}_R}\|g_P\|^2 + \mathbb{E}_{\mathcal{P}_R}\|g_{I_1}\|^2 + \mathbb{E}_{\mathcal{P}_R}\|g_{I_2}\|^2 - 2\mathbb{E}_{\mathcal{P}_R}\left(\sum_{i\in I_1\bigcup I_2}g_i^2\right).
\end{aligned}$$

On the other hand,

$$\begin{aligned}
E_1 + E_2 \quad &= \quad 2\mathbb{E}_{\mathcal{P}_R}\|g_P\|^2 + \mathbb{E}_{\mathcal{P}_R}\|g_{I_1}\|^2 + \mathbb{E}_{\mathcal{P}_R}\|g_{I_2}\|^2 - 2\mathbb{E}_{\mathcal{P}_R}\left(\sum_{i\in I_1}g_i^2\right) - 2\mathbb{E}_{\mathcal{P}_R}\left(\sum_{i\in I_2}g_i^2\right) \\
&= \quad 2\mathbb{E}_{\mathcal{P}_R}\|g_P\|^2 + \mathbb{E}_{\mathcal{P}_R}\|g_{I_1}\|^2 + \mathbb{E}_{\mathcal{P}_R}\|g_{I_2}\|^2 - 2\mathbb{E}_{\mathcal{P}_R}\left(\sum_{i\in I_1\bigcup I_2}g_i^2\right),
\end{aligned}$$

together with $\mathbb{E}_{\mathcal{P}_R}\|g_P\|^2 \geq 0$ implies $E \leq E_1 + E_2$. $\qquad\square$

To measure the compression error due to the compressed gradient $\mathcal{C}_{\mathcal{P}_R,\delta}(g)$ with respect to the full gradient vector $g$, by Lemma 9, there exists an $\alpha \in \mathbb{R}^+$ such that the total expected compression error:

$$\mathbb{E}\|g - \mathcal{C}_{\mathcal{P}_R,\delta}(g)\|^2 = E + \sum_{i\in P^c}g_i^2 \leq \alpha\|g\|^2. \tag{8}$$

But there is no guarantee that $\alpha \in [0, 1)$. On the other hand, by Remark 1 we have:

$$\mathbb{E}\|g - \mathcal{C}_{\mathcal{P}_R,\delta}(g)\|^2 = \mathbb{E}\|g - \text{Random}r(g)\|^2 \leq (1 - \frac{r}{d})\|g\|^2, \tag{9}$$

which guarantees $\mathcal{C}_{\mathcal{P}_R,\delta}$ to be a $\delta$-compressor. Additionally, the sparsifier, $\mathcal{C}_{\mathcal{P}_R,\delta}(g)$ is an hybrid sprasifier—It has some attributes of the original sparsifier, $\mathcal{C}_\delta$, but we are unsure which $k_1$ and $k_2$ random elements are selected via policy $\mathcal{P}_R$. If $k_1 = 0$, then $\mathcal{C}_{\mathcal{P}_R,\delta}(g)$ is Random-$r$ [69] sparsifier. If $k_2 = 0$, then $\mathcal{C}_{\mathcal{P}_R,\delta}(g)$ is $\mathcal{C}_\delta$ sparsifier. Furthermore, if $\mathcal{C}_\delta$ is Top-$r$, then $\mathcal{C}_{\mathcal{P}_R,\delta}(g)$ is similar to hybrid random-Top-$r$ sparsifier by Elibol et al. [24]. If $\mathcal{C}_\delta$ is Top-$r$, $k_1 \leq r, k_2 = 0$, then it is random-Top-$k_1$ sparsifier of Barnes et al. [9].

For inherently sparse vectors $g$, with $\mathcal{C}_\delta = I_d$, we have $\mathcal{C}_{\mathcal{P}_R,I_d}(g) = g_{I_1}$ and Lemma 9 holds. Moreover, by (8) we have:

$$\mathbb{E}\|g - \mathcal{C}_{\mathcal{P}_R,I_d}(g)\|^2 = (1 - \frac{k_1}{r})\|g\|^2. \tag{10}$$

That is, the policy creates a *lossy* compression with compression factor as good as a Random-$k_1$ compressor.

---

**Algorithm 1** Construct bloom filter for policy BF-P2

---

**Input:** Bloom filter $\mathcal{B}$ of size $m$, $k$-hash functions $h_i$, gradient dimensionality $d$, empty set $P$, empty
       conflict sets $C_j$, empty set $\tilde{S}$, target number of decompressed indices $r$
**Output:** A set of decompressed indices $\tilde{S}$
**for** $i = 1$ *to* $d$ **do** /* all $d$ elements of gradient $\tilde{g} \in \mathbb{R}^d$ */
     **if** $i \in \mathcal{B}$ **then** insert $i$ in $P$
**for** *each* $x \in P$ **do**
     **for** $i = 1$ *to* $k$ **do** insert $x$ in $C_{h_i(x)}$
Sort conflict-sets in $C$ by their sizes in ascending order
**while** $size(\tilde{S}) < r$ **do**
     **for** *each* $C_j \in C$ **do**
         **if** $|C_j| = 1$ **then** Insert $C_j$ in $\tilde{S}$; Remove $C_j$ from $C$
         **else**
             Remove from $C_j$ items that exist in $\tilde{S}$ Insert into $\tilde{S}$ a random item from $C_j$
**return** $\tilde{S}$

---

### B.5 Algorithmic details of conflict set policy: Policy P2

In the following, we explain the conflict set policy, P2. Pseudocode in Algorithm 1 presents the details. Lines 1-2 construct set $P$, i.e., the union of the true and false positive responses of $\mathcal{B}$. Lines 3-4 re-hash the items of $P$ into $\mathcal{B}$ to construct conflict sets $C_{1...j}$, where $j$ is equal to the number of "1"s in $\mathcal{B}$. For lack of better information, we assume that the true positives are uniformly distributed across the conflict sets; therefore, the probability of drawing a true positive out of a smaller conflict set is higher. To prioritize such sets, Line 5 sorts $C_{1...j}$ in ascending size order. Then, lines 6-11 repeatedly draw items out of the conflict sets until the size of $\tilde{S}$ reaches our target size $r$. If a set $C_j$ is initially a singleton, its item is a true positive; thus it is added to $\tilde{S}$. Else, we remove from $C_j$ any items that already exist in $\tilde{S}$ (observe there may be duplicates among conflict sets), and add randomly a remaining item to $\tilde{S}$.

## C Approximation error due to polynomial fit

We discuss the missing details of value fitting (Section 5) in the following and then provide the compression error from value fitting in Section C.1. The first result is concerning the knot selections.

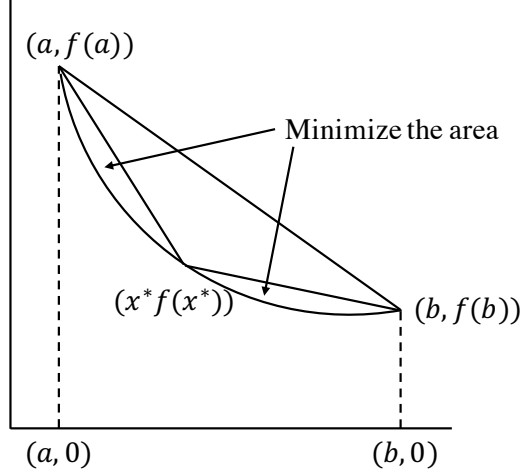

Figure 14: Knot selection for $y = f(x)$ on the interval $[a, b]$.

**Lemma 10.** *(Knot selection) If $f$ is a differentiable convex function on $[a, b]$, then the point $x^* \in (a, b)$ that gives the best approximation to $f$ using line segments from $(a, f(a))$ to $(x^*, f(x^*))$, and then*

*from $(x^*, f(x^*))$ to $(b, f(b))$ is the same point $x^+$ that maximizes the difference between $f$ and the line segment connecting $(a, f(a))$ to $(b, f(b))$.*

*Proof.* The goal is to find $x^*$ such that it minimizes $\int_a^{x^*} \left[ \frac{f(x^*)-f(a)}{x^*-a}(x-a) + f(a) - f(x) \right] dx + \int_{x^*}^b \left[ \frac{f(b)-f(x^*)}{b-x^*}(x-b) + f(b) - f(x) \right] dx$ among all choices of $x^* \in (a, b)$; see Figure 14.

Let

$$F(t) = \int_a^t \left[ \frac{f(t)-f(a)}{t-a}(x-a) + f(a) - f(x) \right] dx + \int_t^b \left[ \frac{f(b)-f(t)}{b-t}(x-b) + f(b) - f(x) \right] dx.$$

Then

$$F'(t) = \int_a^t \frac{d}{dt} \left[ \frac{f(t)-f(a)}{t-a}(x-a) \right]_t dx + \int_t^b \frac{d}{dt} \left[ \frac{f(b)-f(t)}{b-t}(x-b) \right]_t dx$$

$$= = \frac{1}{2}[f'(t)(b-a) - (f(b) - f(a))].$$

So, the critical point satisfies

$$f'(t)(b-a) - (f(b) - f(a)) = 0,$$
$$\text{that is,} \qquad f'(t) = \frac{f(b)-f(a)}{b-a}. \tag{11}$$

Now consider,

$$A(x) = \max_{x\in[a,b]} \left[ \frac{f(b) - f(a)}{b - a}(x - a) + f(a) - f(x) \right].$$

It is easy to see that the critical point of the function $A(x)$ alone satisfies $f'(x) = \frac{f(b)-f(a)}{b-a}$, which is the same solution as in (11). $\qquad\square$

**Polynomial regression.** As mentioned in Section 5, over each sorted segment, we apply a polynomial regression. In our experiments, we usually take the degree of the polynomial as 5. The following result concerns piece-wise constant approximation.

**Lemma 11.** *[21] (Error of piece-wise constant fit) For $\mathcal{C}_S(g) \in C([1, d])$ and $M > 0$ the following are equivalent:*
*(i) $\mathrm{Var}_{[1,d]}(\mathcal{C}_S(g)) \leq M$ and (ii) $\|s - \mathcal{C}_S(g)\|_\infty \leq \frac{M}{2p+2}$, for some $s \in \mathcal{S}_p^0$— the set of piece-wise constant splines with $p$ knots.*

Proof of the above Lemma follows from [21].

*Remark* 4. We can also give a heuristic to calculate $p$ from Lemma 11. First, calculate $M = |(\mathcal{C}_S(g)[1] - \mathcal{C}_S(g)[2]) - (\mathcal{C}_S(g)[d - 1] - \mathcal{C}_S(g)[d])|$. By considering the error bound $\frac{M}{2p+2}$ as a function of $p$, we can find closed-form solution for $p$ as $p = \lceil \frac{M}{\sqrt{2}} - 1 \rceil$.

However, for piece-wise linear fit, we propose the following result with an explicit constant.

**Lemma 12.** *(Error of piece-wise linear fit with explicit constants) For $\mathcal{C}_S(g) \in C^1([1, d])$ with $\mathrm{Var}_{[1,d]}(\mathcal{C}_S'(g)) \leq M$, we have $\|s - \mathcal{C}_S(g)\|_\infty \leq \frac{2M}{p^2}$, for some $s \in \mathcal{S}_p^1$— the set of piece-wise linear splines with $p$ knots.*

*Proof.* Let $p' = \lceil p/2 \rceil$, the integer part of $p/2$, and let $s^0 \in S_p^0$ satisfy $|\mathcal{C}_S'(g)(t) - s^0(t)| \leq \frac{M}{2p'+2}$, as guaranteed by Lemma 11. Denote $V^1 := \int_1^T |\mathcal{C}_S'(g)(t) - s^0(t)|dt$. Choose $p'$ knots $\{t_j\}$ in $(1, d)$ such that $\int_{t_j}^{t_{j+1}} |\mathcal{C}_S'(g) - s^0(t)|dt = \frac{V^1}{p'+1}$. Define $s(x) = \int_{t_j}^x s^0(t)dt + f(t_j)$ for $x \in [t_j, t_{j+1})$, $j = 0, 1, ..., p'$. Then $s$ is a piece-wise linear spline function with possible knots at $2p' \leq p$ points such that, for $x \in [t_j, t_{j+1})$, $|\mathcal{C}_S(g)(x) - s(x)| \leq \int_{t_j}^x |\mathcal{C}_S'(g)(t) - s^0(t)|dt \leq \int_{t_j}^{t_{j+1}} |\mathcal{C}_S(g)(t) - s^0(t)|dt = \frac{2M}{(2p'+2)^2} \leq \frac{2M}{p^2}$, which implies the result. $\qquad\square$

*Remark* 5. Let $s \in \mathcal{S}_p^1$. Denote $\hat{\sigma} := \|s - \mathcal{C}_S(g)\|$. Then

$$\hat{\sigma} := \|s - \mathcal{C}_S(g)\| \leq \sqrt{d}\|s - \mathcal{C}_S(g)\|_\infty \overset{\text{By Lemma 12}}{\leq} \frac{2\sqrt{d}M}{p^2}.$$

## C.1 Compression error from value fitting

Now we are set to discuss about the compression error from value fitting. Let $\hat{\mathcal{C}}(g)$ be the approximation of the sparse vector, $\mathcal{C}_\delta(g)$ resulted from a $\delta$-compressor. In the intermediate step, we consider the sparse vector, $\mathcal{C}_S(g)$ that has the components of $\mathcal{C}_\delta(g)$ arranged in descending order of magnitude. Let $s \in \mathcal{S}_p^1$ be the approximation of $\mathcal{C}_S(g)$. Assume no orthogonality and let the angle between $\mathcal{C}_\delta(g)$ and $\mathcal{C}_S(g)$ be $0 \le \theta < \pi/2$, and the angle between $s$ and $\hat{\mathcal{C}}(g)$ be $0 \le \theta' < \pi/2$. We aim to calculate the bound on $\mathbb{E}\|\hat{\mathcal{C}}(g)\|$, (where $\hat{\mathcal{C}}(g)$ is considered to be iteration and worker agnostic) and the Lemma follows.

**Lemma 13.** *(i) If $\mathcal{C}_\delta(g)$ is unbiased, that is, $\mathbb{E}(\mathcal{C}_\delta(g)) = g$ then*

$$\mathbb{E}\|\hat{\mathcal{C}}(g)\|^2 \le 2(2-\delta)(21 - 4\cos\theta - 16\cos\theta')\|g\|^2 + (5 - 4\cos\theta')\frac{32dM^2}{p^4}. \tag{12}$$

*(ii) If $\mathcal{C}_\delta(g)$ is biased, that is, $\mathbb{E}(\mathcal{C}_\delta(g)) \ne g$, then*

$$\mathbb{E}\|\hat{\mathcal{C}}(g)\|^2 \le 2(3-2\delta)\|g\|^2 + 16(2-\delta)(5 - \cos\theta - 4\cos\theta')\|g\|^2 + (5 - 4\cos\theta')\frac{32dM^2}{p^4}. \tag{13}$$

*Proof.* We have

$$\mathbb{E}\|\hat{\mathcal{C}}(g)\|^2 = \mathbb{E}\|\hat{\mathcal{C}}(g) - g + g\|^2 \overset{\text{By (3)}}{\le} 2\mathbb{E}\|\hat{\mathcal{C}}(g) - g\|^2 + 2\|g\|^2.$$

**Case 1:** Consider $\mathcal{C}_\delta(g)$ be unbiased, that is, $\mathbb{E}(\mathcal{C}_\delta(g)) = g$ . Therefore,

$$\begin{aligned}
\mathbb{E}\|g - \hat{\mathcal{C}}(g)\|^2 &= \mathbb{E}\|g - \mathcal{C}_\delta(g) + \mathcal{C}_\delta(g) - \hat{\mathcal{C}}(g)\|^2 \\
&\overset{\mathbb{E}(\mathcal{C}_\delta(g))=g}{=} \mathbb{E}\|g - \mathcal{C}_\delta(g)\|^2 + \mathbb{E}\|\mathcal{C}_\delta(g) - \hat{\mathcal{C}}(g)\|^2 \\
&\overset{\text{By (1)}}{\le} (1 - \delta)\|g\|^2 + \mathbb{E}\|\mathcal{C}_\delta(g) - \hat{\mathcal{C}}(g)\|^2.
\end{aligned}$$

Further we need to bound $\mathbb{E}\|\mathcal{C}_\delta(g) - \hat{\mathcal{C}}(g)\|^2$. We have

$$\begin{aligned}
\mathbb{E}\|\mathcal{C}_\delta(g) - \hat{\mathcal{C}}(g)\|^2 &= \mathbb{E}\|\mathcal{C}_\delta(g) - \mathcal{C}_S(g) + \mathcal{C}_S(g) - \hat{\mathcal{C}}(g)\|^2 \\
&\overset{\text{By (3)}}{\le} 2\mathbb{E}\|\mathcal{C}_\delta(g) - \mathcal{C}_S(g)\|^2 + 2\mathbb{E}\|\mathcal{C}_S(g) - \hat{\mathcal{C}}(g)\|^2.
\end{aligned}$$

If the angle between $\mathcal{C}_\delta(g)$ and $\mathcal{C}_S(g)$ be $0 \le \theta < \pi/2$, then

$$\mathbb{E}\|\mathcal{C}_\delta(g) - \mathcal{C}_S(g)\|^2 \overset{\text{Lemma 4}}{=} 2(1 - \cos\theta)\mathbb{E}\|\mathcal{C}_\delta(g)\|^2 \overset{\text{Lemma 5}(i)}{\le} 2(1 - \cos\theta)(2 - \delta)\|g\|^2. \tag{14}$$

We pause here and quantify: $\|\mathcal{C}_S(g) - \hat{\mathcal{C}}(g)\|^2$. Let $s \in \mathcal{S}_p^1$ be the approximation of $\mathcal{C}_S(g)$. Then

$$\begin{aligned}
\mathbb{E}\|\mathcal{C}_S(g) - \hat{\mathcal{C}}(g)\|^2 &= \mathbb{E}\|\mathcal{C}_S(g) - s + s - \hat{\mathcal{C}}(g)\|^2 \\
&\overset{\text{By (3)}}{\le} 2\mathbb{E}\|\mathcal{C}_S(g) - s\|^2 + 2\mathbb{E}\|s - \hat{\mathcal{C}}(g)\|^2 \\
&\overset{\text{By Remark 5}}{\le} \frac{8dM^2}{p^4} + 2\mathbb{E}\|s - \hat{\mathcal{C}}(g)\|^2.
\end{aligned} \tag{15}$$

Similarly, if the angle between $s$ and $\hat{\mathcal{C}}(g)$ be $0 \le \theta' < \pi/2$, then

$$\begin{aligned}
\mathbb{E}\|s - \hat{\mathcal{C}}(g)\|^2 &\overset{\text{By Lemma 4}}{=} 2(1 - \cos\theta')\mathbb{E}\|s\|^2 \\
&\le 2(1 - \cos\theta')\mathbb{E}\|s - \mathcal{C}_S(g) + \mathcal{C}_S(g)\|^2 \\
&\overset{\text{By (3)}}{\le} 4(1 - \cos\theta')\|s - \mathcal{C}_S(g)\|^2 + 4(1 - \cos\theta')\mathbb{E}\|\mathcal{C}_S(g)\|^2 \\
&\overset{\mathbb{E}\|\mathcal{C}_\delta(g)\|^2 = \mathbb{E}\|\mathcal{C}_S(g)\|^2}{=} (1 - \cos\theta')\frac{16dM^2}{p^4} + 4(1 - \cos\theta')\mathbb{E}\|\mathcal{C}_\delta(g)\|^2 \\
&\overset{\text{By Lemma 5}(i)}{\le} (1 - \cos\theta')\frac{16dM^2}{p^4} + 4(1 - \cos\theta')(2 - \delta)\|g\|^2.
\end{aligned} \tag{16}$$

Therefore,

$$\mathbb{E}\|\mathcal{C}_\delta(g) - \hat{\mathcal{C}}(g)\|^2 \leq 2\mathbb{E}\|\mathcal{C}_\delta(g) - \mathcal{C}_S(g)\|^2 + 2\mathbb{E}\|\mathcal{C}_S(g) - \hat{\mathcal{C}}(g)\|^2$$

$$\leq 4(1 - \cos\theta)(2 - \delta)\|g\|^2 + \frac{16dM^2}{p^4} + \frac{64dM^2}{p^4}(1 - \cos\theta')$$

$$+ 16(1 - \cos\theta')(2 - \delta)\|g\|^2.$$

Combining all together we have

$$\mathbb{E}\|\hat{\mathcal{C}}(g)\|^2$$

$$\leq 2(2 - \delta)\|g\|^2 + 8(1 - \cos\theta)(2 - \delta)\|g\|^2 + \frac{32dM^2}{p^4} + \frac{128dM^2}{p^4}(1 - \cos\theta')$$

$$+ 32(1 - \cos\theta')(2 - \delta)\|g\|^2.$$

Arranging the terms, we get the result.

**Case 2:** If $\mathcal{C}_\delta(g)$ be biased, that is, $\mathbb{E}(\mathcal{C}_\delta(g)) \neq g$ then

$$\mathbb{E}\|g - \hat{\mathcal{C}}(g)\|^2 \leq 2\mathbb{E}\|g - \mathcal{C}_\delta(g)\|^2 + 2\mathbb{E}\|\mathcal{C}_\delta(g) - \hat{\mathcal{C}}(g)\|^2$$

$$\overset{\text{By (1)}}{\leq} 2(1 - \delta)\|g\|^2 + 2\mathbb{E}\|\mathcal{C}_\delta(g) - \hat{\mathcal{C}}(g)\|^2.$$

For biased compressor $\mathcal{C}_\delta(g)$, by using Lemma 5 $(ii)$ we have

$$\mathbb{E}\|\mathcal{C}_\delta(g) - \mathcal{C}_S(g)\|^2 \overset{\text{By Lemma 4}}{=} 2(1 - \cos\theta)\mathbb{E}\|\mathcal{C}_\delta(g)\|^2$$

$$\overset{\text{By Lemma 5}(ii)}{\leq} 4(1 - \cos\theta)(2 - \delta)\|g\|^2. \tag{17}$$

and

$$\mathbb{E}\|s - \hat{\mathcal{C}}(g)\|^2 \overset{\text{By Lemma 4}}{=} 4(1 - \cos\theta')\|s - C_S(g)\|^2 + 4(1 - \cos\theta')\mathbb{E}\|\mathcal{C}_\delta(g)\|^2$$

$$\overset{\text{By Lemma 5}(ii)}{\leq} (1 - \cos\theta')\frac{16dM^2}{p^4} + 8(1 - \cos\theta')(2 - \delta)\|g\|^2. \tag{18}$$

Finally,

$$\mathbb{E}\|\mathcal{C}_\delta(g) - \hat{\mathcal{C}}(g)\|^2 \leq 2\mathbb{E}\|\mathcal{C}_\delta(g) - \mathcal{C}_S(g)\|^2 + 2\mathbb{E}\|\mathcal{C}_S(g) - \hat{\mathcal{C}}(g)\|^2$$

$$\leq 8(1 - \cos\theta)(2 - \delta)\|g\|^2 + \frac{16dM^2}{p^4} + (1 - \cos\theta')\frac{64dM^2}{p^4}$$

$$+ 32(1 - \cos\theta')(2 - \delta)\|g\|^2.$$

Combining all together we have

$$\mathbb{E}\|\hat{\mathcal{C}}(g)\|^2$$

$$\leq 2\mathbb{E}\|\hat{\mathcal{C}}(g) - g\|^2 + 2\|g\|^2$$

$$\leq 2(3 - 2\delta)\|g\|^2 + 16(1 - \cos\theta)(2 - \delta)\|g\|^2 + \frac{32dM^2}{p^4} + (1 - \cos\theta')\frac{128dM^2}{p^4}$$

$$+ 64(1 - \cos\theta')(2 - \delta)\|g\|^2.$$

Arranging the terms, we get the result. $\square$

## D  Compression error from combined index and value fitting

In this section, we discuss about the compression error from joint index and value fitting. Let $\bar{\mathcal{C}}(g)$ be the sparse approximation of the vector, $g$ after sparse vector, $\mathcal{C}_\delta(g)$ resulted from a $\delta$-compressor, goes through consequent index compression via $\mathcal{P}_R$,[2] and value compression via piecewise polynomial fit.

---

[2]We give the result by using random policy, $\mathcal{P}_R$. Also, similar bounds hold for deterministic policy. For $P_0$, the quantity, $\frac{r}{d}$ in the proofs will be replaced by $\beta$ with $0 < \beta \leq 1$.

Let $\mathcal{C}_{\mathcal{P}_R,\delta}(g)$ be the sparse vector whose indices are resulted from policy $\mathcal{P}_R$ applied to $\mathcal{C}_\delta(g)$. In the intermediate step, we consider the sparse vector, $\mathcal{C}_S(g)$ that has the components of $\mathcal{C}_{\mathcal{P}_R,\delta}(g)$ arranged in descending order of magnitude. Let $s \in \mathcal{S}_p^1$ be the approximation of $\mathcal{C}_S(g)$. Let the angle between $\mathcal{C}_{\mathcal{P}_R,\delta}(g)$ and $\mathcal{C}_S(g)$ be $0 \le \theta < \pi/2$, and the angle between $s$ and $\bar{\mathcal{C}}(g)$ be $0 \le \theta' < \pi/2$. We aim to calculate the bound on $\mathbb{E}\|\bar{\mathcal{C}}(g)\|$, (where $\bar{\mathcal{C}}(g)$ is considered to be iteration and worker agnostic) and the Lemma follows.

**Lemma 14.** *(i) If $\mathcal{C}_\delta(g)$ is unbiased, that is, $\mathbb{E}(\mathcal{C}_\delta(g)) = g$ then*

$$
\begin{aligned}
\mathbb{E}\|\bar{\mathcal{C}}(g)\|^2 \;\le\;\; & 2(2-\delta)\|g\|^2 + 4(2 - \tfrac{r}{d} - \delta)\|g\|^2 + 32(1-\cos\theta)(2 - \tfrac{r}{d})\|g\|^2 + \frac{64dM^2}{p^4} \\
& + (1-\cos\theta')\frac{256dM^2}{p^4} + 64(1-\cos\theta')(2-\delta)\|g\|^2.
\end{aligned} \tag{19}
$$

*(ii) If $\mathcal{C}_\delta(g)$ is biased, that is, $\mathbb{E}(\mathcal{C}_\delta(g)) \ne g$, then*

$$
\begin{aligned}
\mathbb{E}\|\bar{\mathcal{C}}(g)\|^2 \;\le\;\; & 2(3-\delta)\|g\|^2 + 16(2 - \tfrac{r}{d} - \delta)\|g\|^2 + 32(1-\cos\theta)(2 - \tfrac{r}{d})\|g\|^2 + \frac{64dM^2}{p^4} \\
& + (1-\cos\theta')\frac{256dM^2}{p^4} + 128(1-\cos\theta')(2-\delta)\|g\|^2.
\end{aligned} \tag{20}
$$

*Proof.* We have

$$
\mathbb{E}\|\bar{\mathcal{C}}(g)\|^2 = \mathbb{E}\|\bar{\mathcal{C}}(g) - g + g\|^2 \overset{\text{By (3)}}{\le} 2\mathbb{E}\|\bar{\mathcal{C}}(g) - g\|^2 + 2\|g\|^2.
$$

**Case 1:** Consider $\mathcal{C}_\delta(g)$ be unbiased, that is, $\mathbb{E}(\mathcal{C}_\delta(g)) = g$ . Therefore,

$$
\begin{aligned}
\mathbb{E}\|g - \bar{\mathcal{C}}(g)\|^2 \;&=\; \mathbb{E}\|g - \mathcal{C}_\delta(g) + \mathcal{C}_\delta(g) - \bar{\mathcal{C}}(g)\|^2 \\
&\overset{\mathbb{E}(\mathcal{C}_\delta(g))=g}{=}\; \mathbb{E}\|g - \mathcal{C}_\delta(g)\|^2 + \mathbb{E}\|\mathcal{C}_\delta(g) - \bar{\mathcal{C}}(g)\|^2 \\
&\overset{\text{By (1)}}{\le}\; (1-\delta)\|g\|^2 + \mathbb{E}\|\mathcal{C}_\delta(g) - \bar{\mathcal{C}}(g)\|^2.
\end{aligned}
$$

Further, we need to bound $\mathbb{E}\|\mathcal{C}_\delta(g) - \bar{\mathcal{C}}(g)\|^2$. We have

$$
\begin{aligned}
\mathbb{E}\|\mathcal{C}_\delta(g) - \bar{\mathcal{C}}(g)\|^2 \;&=\; \mathbb{E}\|\mathcal{C}_\delta(g) - \mathcal{C}_{\mathcal{P}_R,\delta}(g) + \mathcal{C}_{\mathcal{P}_R,\delta}(g) - \bar{\mathcal{C}}(g)\|^2 \\
&\overset{\text{By (3)}}{\le}\; 2\mathbb{E}\|\mathcal{C}_\delta(g) - \mathcal{C}_{\mathcal{P}_R,\delta}(g)\|^2 + 2\mathbb{E}\|\mathcal{C}_{\mathcal{P}_R,\delta}(g) - \bar{\mathcal{C}}(g)\|^2 \\
&=\; 2\mathbb{E}\|g - \mathcal{C}_{\mathcal{P}_R,\delta}(g) - (g - \mathcal{C}_\delta(g))\|^2 + 2\mathbb{E}\|\mathcal{C}_{\mathcal{P}_R,\delta}(g) - \bar{\mathcal{C}}(g)\|^2 \\
&\overset{\text{By (9) (3),and (1)}}{\le}\; 2(1-\tfrac{r}{d})\|g\|^2 + 2(1-\delta)\|g\|^2 + 2\mathbb{E}\|\mathcal{C}_{\mathcal{P}_R,\delta}(g) - \bar{\mathcal{C}}(g)\|^2 \\
&=\; 2(2-\tfrac{r}{d}-\delta)\|g\|^2 + 2\mathbb{E}\|\mathcal{C}_{\mathcal{P}_R,\delta}(g) - \bar{\mathcal{C}}(g)\|^2.
\end{aligned}
$$

Now,

$$
\begin{aligned}
\mathbb{E}\|\mathcal{C}_{\mathcal{P}_R,\delta}(g) - \bar{\mathcal{C}}(g)\|^2 \;&=\; \mathbb{E}\|\mathcal{C}_{\mathcal{P}_R,\delta}(g) - \mathcal{C}_S(g) + \mathcal{C}_S(g) - \bar{\mathcal{C}}(g)\|^2 \\
&\overset{\text{By (3)}}{\le}\; 2\mathbb{E}\|\mathcal{C}_{\mathcal{P}_R,\delta}(g) - \mathcal{C}_S(g)\|^2 + 2\mathbb{E}\|\mathcal{C}_S(g) - \bar{\mathcal{C}}(g)\|^2.
\end{aligned}
$$

If the angle between $\mathcal{C}_{\mathcal{P}_R,\delta}(g)$ and $\mathcal{C}_S(g)$ be $0 \le \theta < \pi/2$, then

$$
\mathbb{E}\|\mathcal{C}_{\mathcal{P}_R,\delta}(g) - \mathcal{C}_S(g)\|^2 \overset{\text{Lemma 4}}{=} 2(1-\cos\theta)\mathbb{E}\|\mathcal{C}_{\mathcal{P}_R,\delta}(g)\|^2 \overset{\text{By (9) and (3)}}{\le} 2(1-\cos\theta)(2 - \tfrac{r}{d})\|g\|^2. \tag{21}
$$

We pause here and quantify: $\|\mathcal{C}_S(g) - \bar{\mathcal{C}}(g)\|^2$. Let $s \in \mathcal{S}_p^1$ be the approximation of $\mathcal{C}_S(g)$. Then

$$
\begin{aligned}
\mathbb{E}\|\mathcal{C}_S(g) - \bar{\mathcal{C}}(g)\|^2 \;&=\; \mathbb{E}\|\mathcal{C}_S(g) - s + s - \bar{\mathcal{C}}(g)\|^2 \\
&\overset{\text{By (3)}}{\le}\; 2\mathbb{E}\|\mathcal{C}_S(g) - s\|^2 + 2\mathbb{E}\|s - \bar{\mathcal{C}}(g)\|^2 \\
&\overset{\text{By Remark 5}}{\le}\; \frac{8dM^2}{p^4} + 2\mathbb{E}\|s - \bar{\mathcal{C}}(g)\|^2.
\end{aligned} \tag{22}
$$

Similarly, if the angle between $s$ and $\bar{\mathcal{C}}(g)$ be $0 \leq \theta' < \pi/2$, then

$$
\begin{aligned}
\mathbb{E}\|s - \bar{\mathcal{C}}(g)\|^2 &\overset{\text{By Lemma 4}}{=} 2(1 - \cos\theta')\mathbb{E}\|s\|^2 \\
&\leq 2(1 - \cos\theta')\mathbb{E}\|s - \mathcal{C}_S(g) + \mathcal{C}_S(g)\|^2 \\
&\overset{\text{By (3)}}{\leq} 4(1 - \cos\theta')\|s - \mathcal{C}_S(g)\|^2 + 4(1 - \cos\theta')\mathbb{E}\|\mathcal{C}_S(g)\|^2 \\
&\overset{\mathbb{E}\|\mathcal{C}_{\mathcal{P}_R,\delta}(g)\|^2 = \mathbb{E}\|\mathcal{C}_S(g)\|^2}{=} (1 - \cos\theta')\frac{16dM^2}{p^4} + 4(1 - \cos\theta')\mathbb{E}\|\mathcal{C}_{\mathcal{P}_R,\delta}(g)\|^2 \\
&\overset{\text{By Lemma 5}(i)}{\leq} (1 - \cos\theta')\frac{16dM^2}{p^4} + 4(1 - \cos\theta')(2 - \frac{r}{d})\|g\|^2. \quad (23)
\end{aligned}
$$

Therefore,

$$
\begin{aligned}
&\mathbb{E}\|\mathcal{C}_{\mathcal{P}_R,\delta}(g) - \bar{\mathcal{C}}(g)\|^2 \\
\leq\ & 2\mathbb{E}\|\mathcal{C}_{\mathcal{P}_R,\delta}(g) - \mathcal{C}_S(g)\|^2 + 2\mathbb{E}\|\mathcal{C}_S(g) - \bar{\mathcal{C}}(g)\|^2 \\
\leq\ & 4(1 - \cos\theta)(2 - \frac{r}{d})\|g\|^2 + \frac{16dM^2}{p^4} + \frac{64dM^2}{p^4}(1 - \cos\theta') + 16(1 - \cos\theta')(2 - \frac{r}{d})\|g\|^2.
\end{aligned}
$$

Combining all together we have

$$
\begin{aligned}
&\mathbb{E}\|\bar{\mathcal{C}}(g)\|^2 \\
\leq\ & 2(2 - \delta)\|g\|^2 + 4(2 - \frac{r}{d} - \delta)\|g\|^2 + 16(1 - \cos\theta')(2 - \frac{r}{d})\|g\|^2 + \frac{64dM^2}{p^4} \\
& + \frac{256dM^2}{p^4}(1 - \cos\theta') + 64(1 - \cos\theta')(2 - \delta)\|g\|^2.
\end{aligned}
$$

Arranging the terms, we get the result.

**Case 2:** If $\mathcal{C}_\delta(g)$ be biased, that is, $\mathbb{E}(\mathcal{C}_\delta(g)) \neq g$ then

$$
\begin{aligned}
\mathbb{E}\|g - \bar{\mathcal{C}}(g)\|^2 &\leq 2\mathbb{E}\|g - \mathcal{C}_\delta(g)\|^2 + 2\mathbb{E}\|\mathcal{C}_\delta(g) - \bar{\mathcal{C}}(g)\|^2 \\
&\overset{\text{By (1)}}{\leq} 2(1 - \delta)\|g\|^2 + 2\mathbb{E}\|\mathcal{C}_\delta(g) - \bar{\mathcal{C}}(g)\|^2.
\end{aligned}
$$

Further, we have

$$
\begin{aligned}
\mathbb{E}\|\mathcal{C}_\delta(g) - \bar{\mathcal{C}}(g)\|^2 &= \mathbb{E}\|\mathcal{C}_\delta(g) - \mathcal{C}_{\mathcal{P}_R,\delta}(g) + \mathcal{C}_{\mathcal{P}_R,\delta}(g) - \bar{\mathcal{C}}(g)\|^2 \\
&\overset{\text{By (3)}}{\leq} 2\mathbb{E}\|\mathcal{C}_\delta(g) - \mathcal{C}_{\mathcal{P}_R,\delta}(g)\|^2 + 2\mathbb{E}\|\mathcal{C}_{\mathcal{P}_R,\delta}(g) - \bar{\mathcal{C}}(g)\|^2 \\
&= 2\mathbb{E}\|g - \mathcal{C}_{\mathcal{P}_R,\delta}(g) - (g - \mathcal{C}_\delta(g))\|^2 + 2\mathbb{E}\|\mathcal{C}_{\mathcal{P}_R,\delta}(g) - \bar{\mathcal{C}}(g)\|^2 \\
&\overset{\text{By (9) (3),and (1)}}{\leq} 4(1 - \frac{r}{d})\|g\|^2 + 4(1 - \delta)\|g\|^2 + 2\mathbb{E}\|\mathcal{C}_{\mathcal{P}_R,\delta}(g) - \bar{\mathcal{C}}(g)\|^2 \\
&= 4(2 - \frac{r}{d} - \delta)\|g\|^2 + 2\mathbb{E}\|\mathcal{C}_{\mathcal{P}_R,\delta}(g) - \bar{\mathcal{C}}(g)\|^2.
\end{aligned}
$$

For biased compressor $\mathcal{C}_\delta(g)$, by using Lemma 5 $(ii)$ we have

$$
\begin{aligned}
\mathbb{E}\|\mathcal{C}_{\mathcal{P}_R,\delta}(g) - \mathcal{C}_S(g)\|^2 &\overset{\text{By Lemma 4}}{=} 2(1 - \cos\theta)\mathbb{E}\|\mathcal{C}_{\mathcal{P}_R,\delta}(g)\|^2 \\
&\overset{\text{By Lemma 5}(ii)}{\leq} 4(1 - \cos\theta)(2 - \frac{r}{d})\|g\|^2; \quad (24)
\end{aligned}
$$

and

$$
\begin{aligned}
\mathbb{E}\|s - \bar{\mathcal{C}}(g)\|^2 &\overset{\text{By Lemma 4}}{=} 4(1 - \cos\theta')\|s - \mathcal{C}_S(g)\|^2 + 4(1 - \cos\theta')\mathbb{E}\|\mathcal{C}_{\mathcal{P}_R,\delta}(g)\|^2 \\
&\overset{\text{By Lemma 5}(ii)}{\leq} (1 - \cos\theta')\frac{16dM^2}{p^4} + 8(1 - \cos\theta')(2 - \frac{r}{d})\|g\|^2. \quad (25)
\end{aligned}
$$

Finally,

$$
\begin{aligned}
&\mathbb{E}\|\mathcal{C}_{\mathcal{P}_R,\delta}(g) - \bar{\mathcal{C}}(g)\|^2 \\
\leq\ & 2\mathbb{E}\|\mathcal{C}_{\mathcal{P}_R,\delta}(g) - \mathcal{C}_S(g)\|^2 + 2\mathbb{E}\|\mathcal{C}_S(g) - \bar{\mathcal{C}}(g)\|^2 \\
\leq\ & 8(1 - \cos\theta)(2 - \frac{r}{d})\|g\|^2 + \frac{16dM^2}{p^4} + (1 - \cos\theta')\frac{64dM^2}{p^4} + 32(1 - \cos\theta')(2 - \frac{r}{d})\|g\|^2.
\end{aligned}
$$

Combining all together we have

$$
\begin{aligned}
& \mathbb{E}\|\bar{\mathcal{C}}(g)\|^2 \\
\leq\ & 2\mathbb{E}\|\bar{\mathcal{C}}(g) - g\|^2 + 2\|g\|^2 \\
\leq\ & 2(3 - \delta)\|g\|^2 + 16(2 - \frac{r}{d} - \delta)\|g\|^2 + 32(1 - \cos\theta)(2 - \frac{r}{d})\|g\|^2 + \frac{64dM^2}{p^4} \\
& + (1 - \cos\theta')\frac{256dM^2}{p^4} + 128(1 - \cos\theta')(2 - \delta)\|g\|^2.
\end{aligned}
$$

Arranging the terms, we get the result. $\qquad\square$

## D.1 Convergence of distributed compressed SGD without error feedback

We comment on the convergence of compressed distributed SGD without error feedback [23]. We consider the following scenarios:

1. **Approximation by using only value compression.** One can find the bound on compressed aggregated gradient $\tilde{g}_k$ resulting at $k^{\text{th}}$ iteration. Denote $\hat{\mathcal{C}}(g_k^i)$ be the approximation of the sparse vector $\mathcal{C}_\delta(g_k^i)$ resulted from a $\delta$-compressor at $i^{\text{th}}$ worker, at the $k^{\text{th}}$ iteration. Denote the compressed aggregated gradient at $k^{\text{th}}$ iteration to be $\tilde{g}_k := \frac{1}{n}\sum_{i=1}^n \hat{\mathcal{C}}(g_k^i)$. By using Lemma 13, we can find bound on $\mathbb{E}\|\tilde{g}_k\|^2$ is for both biased and unbiased $\mathcal{C}_\delta(g)$. [3]

2. **Approximation by using both value and index compression.** Similarly, denote $\bar{\mathcal{C}}(g_k^i)$ be the approximation of the sparse vector $\mathcal{C}_\delta(g_k^i)$ resulted from a $\delta$-compressor at $i^{\text{th}}$ worker, at the $k^{\text{th}}$ iteration by consequent index compression via $\mathcal{P}_R$, and value compression via piecewise polynomial fit. Denote the compressed aggregated gradient at $k^{\text{th}}$ iteration to be $g_k^\perp := \frac{1}{n}\sum_{i=1}^n \bar{\mathcal{C}}(g_k^i)$. By using Lemma 14, we can find bound $\mathbb{E}\|g_k^\perp\|^2$ for both biased and unbiased $\mathcal{C}_\delta(g)$.

With the above, based on the strong growth condition of stochastic gradients [23, 77], for a lower bounded, Lipschitz smooth, and non-convex loss function $f$, following [23], the distributed SGD with an $\delta$ sparsifier converges, that is, $\min_{k\in[T]} \mathbb{E}(\|\nabla f_k\|^2) \to 0$ as $T \to \infty$. The convergence with error-feedback [45] is a more mathematically involved problem that requires independent investigation, and left for future research.

For the convergence of compressed FedAvg algorithm, we refer to the recent unified analysis in [30] (also see [60, 31]). However, convergence of bidirectional compressed FedAvg with error feedback is an open problem and not the scope of this paper.

## E Implementation details

This section highlights the implementation of different Bloom policies for index compression, polynomial regression for value compression, and joint index and value compression on GPUs and CPUs by using popular deep learning toolkits, TensorFlow and PyTorch (see Table 3).

**Hash-Functions used in Bloom Filter implementation.** We use MurmurHash (MurmurHash3) to construct the hash table in the GPU implementation of the Bloom filter; see Python library https://pypi.org/project/mmh3/. We determine the number of hash functions, $k$, and the length of Bloom filter bit-string, $m$, by using Lemma 3 and Remark 2 in the Appendix.

**Implementation of Bloom Filter on GPUs and CPUs.** We provide an efficient GPU implementation of Bloom filters on PyTorch. During construction, many items can be inserted in parallel without locking, since collisions do not cause inconsistency. Since the domain $[d]$ of the hash functions is finite, we precompute a 2D lookup table $\mathbb{H}^{d,k}$, for each possible input of all hash functions. We store $\mathbb{H}$ in the GPU memory, allowing us to insert items in the Bloom filter using only lookup operations. $\mathbb{H}$ occupies around 1.5MB for ResNet-20 and 1GB for NCF; note that this optimization may not be feasible for very large models. Querying is also implemented in the GPU. If an item $i$ belongs

---

[3]Approximation error from index compression in distributed case, is a simple consequence of (9).

to the Bloom filter, then $\mathcal{B}[h_1(i)] + \mathcal{B}[h_2(i)] + \mathcal{B}[h_3(i)] + \cdots + \mathcal{B}[h_k(1)] == k$. The summation can be executed in parallel with each hash function reduced to a lookup in $\mathbb{H}$. Moreover, many such queries can run concurrently. Although the basic Bloom filter is implemented on GPUs, complex policies, like P2, require programming flexibility. For this reason, we provide CPU implementations on PyTorch, using library pybloomfilter [4]; and on TensorFlow using the C++ extension to create custom operators.

**Implementation of polynomial regression on GPUs and CPUs.** The piece-wise polynomial regression can be solved as a linear problem, once the segments are determined. Our GPU implementation uses Least-Square fitting, which can be trivially expressed with tensor operations. We also provide a CPU implementation using `polyfit` from the NumPy [3] library. We implement the nonlinear double exponential regression on TensorFlow, using tensor operations.

**Combined index and value compression.** To combine Bloom filter-based with curve fitting-based compression, first, observe that neither method is order preserving. Therefore, we need a mapping from the original to the final position of each value. This corresponds to a 1D vector with 1~$d$, where $d$ is the size of sparse gradient. Since now the maximum element in this mapping vector is $d$, we encode their each element using $\lceil \log_2 d \rceil$ bits. For our experiments, this corresponds to 16 bits for ResNet50 and 19 bits for NCF, which is a significant gain compared to the usual int32 format.

**Complexity of the methods.** For each policy of the Bloom filter, if we have $r$ elements and we use $k$ hash functions, then the time and space complexity of each policy is $O(rk) = O(r \log_2(1/\epsilon))$, because the optimal $k = -\frac{\log \epsilon}{\log 2}$, see Remark 2.

We use radix sort which takes $O(d \log_2(d))$, to sort $d$ gradient components. In general, to perform a Top-$r$ selection on CPU on a $d$ dimensional vector, the computational complexity is $O(d \log_2 r)$; for GPUs, other optimized implementations exist [65]. Therefore, to sort these $r$-components further, we require $O(r \log_2(r))$ time, and as $r \ll d$, the total time complexity remains $O(d \log_2 r)$.

For a polynomial fitting with degree $n'$ on each sorted segment with $d_p$ data points, the overall complexity of finding the least-squares solution is $O(d_p n'^2 + n'^3)$. For our application, $n' \ll d_p$, hence, overall complexity is $O(d_p n'^2)$.

We do not use segmentation for nonlinear regression. For nonlinear regression through a double exponential model, $y = ae^{bx} + ce^{dx}$, involves solving a $4 \times 4$ linear system followed by solving a $2 \times 2$ linear system. By using modern solvers (that use Cholesky factorization), the complexity of solving these linear systems are negligible. However, the linear regression in the first system is a proxy to solving integral equations. Therefore, the entries of the coefficient matrix of the first system are approximated via numerical integration (as they cannot be computed by analytical integration) and each of them requires $d$ exact operations, where $d$ is the total number points to be fitted. Similarly, for calculating each entry of the second linear system requires $d$ exact operations.

# F   Additional experimental results

Due to limited space, we were unable to discuss many experimental details as well as many results in Section 6 of the main paper. In this scope, we discuss them in details.

## F.1   Details of the Testbed

**Simulation on local cluster.** For conventional data center experiments, we use 8 dedicated machines with Ubuntu 18.04.2 LTS and Linux v.4.15.0-74, 16-Core Intel Xeon Silver 4112 @ 2.6GHz, 512 GB RAM, one NVIDIA Tesla V100 GPU card with 16 GB on-board memory and 100Gbps network. We deploy CUDA 10.1, TensorFlow 1.14, PyTorch 1.7.1, Horovod 0.21.0, OpenMPI 4.0 and NCCL 2.4.8.

**Realistic federated learning deployment.**   For Federated Learning experiments, we use 57 EC2 instances (g4dn.xlarge) from Amazon Web Service(AWS). The central server is located in Ohio (USA) and 56 clients are spread across 7 different regions globally, each with 8 clients, including Tokyo, Central Canada, Northern California, Seoul, São Paulo, Paris and Oregon. Each client is independently connected to the server with high speed international network. Each instance is

equipped with Ubuntu 16.04.12 LTS, 4-Core Intel(R) Xeon(R) Platinum 8259CL CPU @ 2.50GHz, 16 GB RAM, one NVIDIA Tesla T4 GPU card with 16 GB on-board memory and 128 GB NVMe SSD. We deploy CUDA 11.0, Intel MKL, PyTorch 1.7.1 and MPICH 3.4.1.

### F.2 Implemented methods

We implement DeepReduce [4] as an extension of GRACE [84], a framework that supports many popular sparsification techniques and interfaces with various low-level communication libraries for distributed deep learning. Table 3 presents a summary of the methods we implement.

Table 3: Summary of implementations. $\mathrm{DR}_{idx}^{val}$ denotes instantiation of DeepReduce with $idx$ and $val$ as index and value compression method, respectively.

| Method | Idx | Val | Device | Framework |
|---|:---:|:---:|---|---|
| $\mathrm{DR}_{BF-Naive}^{\varnothing}$ | ✓ | | CPU | TFlow |
| $\mathrm{DR}_{BF-P0}^{\varnothing}$ , $\mathrm{DR}_{BF-P1}^{\varnothing}$ | ✓ | | GPU | PyTorch |
| $\mathrm{DR}_{BF-P0}^{\varnothing}$ , $\mathrm{DR}_{BF-P1}^{\varnothing}$ , $\mathrm{DR}_{BF-P2}^{\varnothing}$ | ✓ | | CPU | TFlow, PyTorch |
| $\mathrm{DR}_{RLE}^{\varnothing}$ | ✓ | | CPU | TFlow, PyTorch |
| $\mathrm{DR}_{Huffman}^{\varnothing}$ | ✓ | | CPU | PyTorch |
| $\mathrm{DR}_{\varnothing}^{Fit-Poly}$ | | ✓ | GPU, CPU | TFlow, PyTorch |
| $\mathrm{DR}_{\varnothing}^{Fit-DExp}$ | | ✓ | GPU | TFlow |
| $\mathrm{DR}_{\varnothing}^{Deflate}$ | | ✓ | CPU | PyTorch |
| $\mathrm{DR}_{\varnothing}^{QSGD}$ | | ✓ | GPU | PyTorch |
| $\mathrm{DR}_{BF-P0}^{Fit-Poly}$ , $\mathrm{DR}_{BF-P1}^{Fit-Poly}$ | ✓ | ✓ | GPU | PyTorch |
| $\mathrm{DR}_{BF-P2}^{QSGD}$ | ✓ | ✓ | GPU | PyTorch |
| 3LC | | ✓ | GPU | TFlow |
| SketchML | | ✓ | CPU | PyTorch |
| SKCompress | ✓ | ✓ | CPU | PyTorch |

**Run Length Encoding (RLE).** Since RLE is a lossless method designed for continuous repetitive symbols, it is not directly applicable to non-repetitive gradient indices. We convert gradient indices into bitmap format, which is a boolean bit string indicating which elements are selected. In this way, RLE can be used to encode the continuous zeros and ones in the bitmap. Note that, the compression rate is highly dependent on the distribution of the indices. That is, RLE is more beneficial if gradient indices contain more continuous integers.

**Huffman Encoding.** The key idea of Huffman Coding is to use fewer bits to represent more frequent symbols. We note that most indices are much smaller than $2^{32}$, and consequently their binary format start with continuous zero bits. Based on this observation, we can use Huffman Coding to compress the binary format of each index to remove the redundancy. The codec is constructed from all possible indices of the target model. (i.e. If the largets gradient size is $d$, then we use $0 \sim d - 1$ for codec construction). The encoding phase contains 2 steps: unpack each 32-bit integer gradient key into Byte format and then encode each index with the pre-defined codec. The decoding phase is just a reversed process.

### F.3 Additional results

**Bloom filter-based index compression: Effect of false positive rate (FPR).** We train ResNet-20 on CIFAR-10 on 8 nodes for 328 epochs and measure the top-1 accuracy and transferred data volume. Our baseline transmits the original uncompressed gradients. To generate sparse gradients, we employ the Top-$r$ [6] and Rand-$r$ [69] sparsifiers; each achieves different accuracy [84]. We vary FPR and measure its effect; smaller FPR corresponds to larger bloom filter. The results for our three index compression policies are shown in Figure 15. Recall (Section 4) that policy BF-P0 transmits extra data for each false positive index. The advantage, as shown in Figure 15a is that accuracy is only marginally affected by FPR, irrespective of the gradient sparsifier (i.e., Top-$r$ or Rand-$r$). The disadvantage is that the amount of transferred data increases with higher FPR; if it is high enough

---
[4]Available at: https://github.com/hangxu0304/DeepReduce

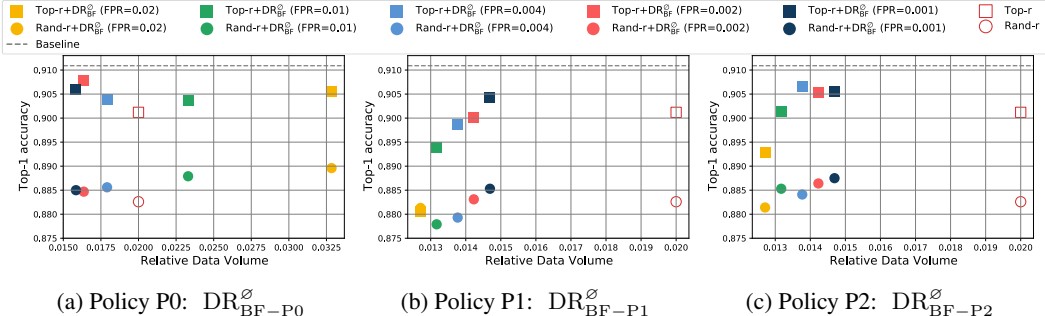

(a) Policy P0: $\mathrm{DR}^{\varnothing}_{\mathrm{BF-P0}}$  (b) Policy P1: $\mathrm{DR}^{\varnothing}_{\mathrm{BF-P1}}$  (c) Policy P2: $\mathrm{DR}^{\varnothing}_{\mathrm{BF-P2}}$

Figure 15: Effect of FPR on top-1 accuracy for the three Bloom filter policies, for ResNet-20 on CIFAR-10. The sparse input gradients were generated by Top-$r$ and Random-$r$ sparsification methods. Data volume is relative to the no-compression baseline.

(e.g., more than 0.004 in our figure), then BF-P0 transfers more data than the sparse input gradient. Policy BF-P1, on the other hand, resolves bloom filter conflicts randomly; as expected, Figure 15b confirms that the amount of transferred data decreases when FPR increases. The trade-off is that accuracy also decreases because more erroneous gradient elements are received. Fortunately, our next policy, BF-P2, improves this issue, as shown in Figure 15c; by resolving conflicts in an informed way.

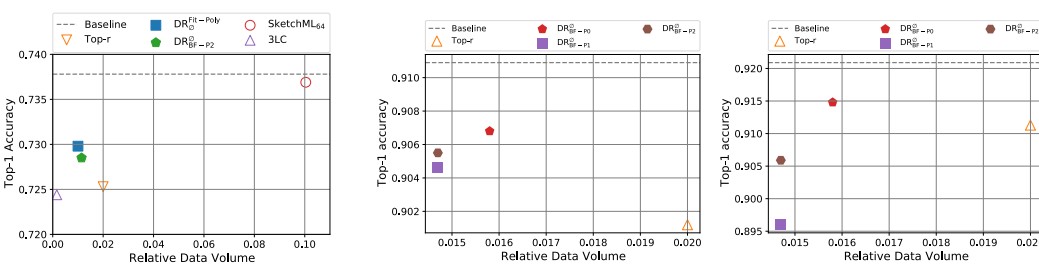

a: ResNet-20 - CIFAR-10    b: DenseNet40-K12 - CIFAR-10

Figure 16: Top-1 test accuracy of ResNet-50 on ImageNet. We compare $\mathrm{DeepReduce}_{\mathrm{idx}}$ and $\mathrm{DeepReduce}_{\mathrm{val}}$ against Top-$r$, $\mathrm{SketchML}_{64}$, 3LC and baseline.

Figure 17: (a) Data volume vs. accuracy of ResNet-20 on CIFAR-10. (b) Data volume vs. accuracy of DensNet40 on CIFAR-10. We compare $\mathrm{DR}^{\varnothing}_{\mathrm{BF}}$ (with P0, P1, P2, Naïve policy) against Top-$r$ and baseline. Ratios of Top-$r$ are 1% for ResNet20, and 0.5% for DenseNet40. FPR is set to 0.001.

**Bloom filter-based index compression: Trade-offs for different polices.** We show the trade-off between accuracy and data volume for different policies of bloom filter in Fig.17. BF-P0 can always maintain the accuracy while sending marginally more data than BF-P1 and BF-P2. BF-P2 has consistently better accuracy than BF-P0 due to the sophisticated conflict-set algorithm.

**DeepReduce on top of the Top-$r$ sparsifier v.s. *stand-alone* compressors.** To contrast with the common practice, we compare DeepReduce against state-of-the-art *stand-alone* gradient compressors [84] that are applied directly on the original gradient. We consider two instantiations of DeepReduce: (*i*) $\mathrm{DR}^{\varnothing}_{\mathrm{BF-P2}}$ that uses Bloom filter index compression with policy P2 and FPR=0.001; and (*ii*) $\mathrm{DR}^{\mathrm{Fit-Poly}}_{\varnothing}$ that uses value compression with polynomial fit. Both operate on the sparse tensor generated by Top-$r$, with $r = 1\%$. We compare against two stand-alone gradient compressors: 3LC [50] with spasification multiplier set to 1, and SketchML [39]; we use $2^6$ quantile buckets, since we opt for best accuracy. For the latter the number of quantile buckets affects accuracy and data volume; for instance, with $2^1$ buckets SketchML achieves only 56.05% Top-1 accuracy. We opt for best accuracy; therefore we use $2^6$ quantile buckets. Memory compensation is enabled for all methods. For this experiment, we employ a much larger benchmark: ResNet-50 on ImageNet. The results are shown in Figure 16, where data volume is relative to the no-compression baseline. Both DeepReduce instantiations provide a good balance between data volume and accuracy, whereas each of the stand-alone methods is biased towards one of the two metrics.

Table 4: Time breakdown and data volume of DeepReduce variants, Top-$r$, and Baseline (FedAvg [55]) in a simulated FL setting. (CLI, SER, S2C and C2S stand for Client, Server, Server-to-Client, and Client-to-Server, respectively.)

| | Avg. Encoding/Decoding Time (s) | | | | Avg. Comm. Time (s) | | Avg. Relative Data Volume | | Test Accuracy |
|---|---|---|---|---|---|---|---|---|---|
| | $CLI_{decode}$ | $CLI_{encode}$ | $SER_{decode}$ | $SER_{encode}$ | S2C | C2S | S2C | C2S | |
| *Baseline | 0 | 0 | 0 | 0 | 1.3980 | 1.3609 | 1.0 | 1.0 | 0.1856 |
| *Top-$r$(0.1) | 0.0044 | 0.0501 | 0.0032 | 0.0258 | 0.3167 | 0.3904 | 0.2033 | 0.2033 | 0.1840 |
| * $DR_{BF-P0}^{\varnothing}$ | 0.0179 | 0.0707 | 0.0183 | 0.0629 | 0.1936 | 0.1957 | 0.1425 | 0.1426 | 0.1841 |
| * $DR_{\varnothing}^{Fit-Poly}$ | 0.0288 | 0.1586 | 0.0183 | 0.1170 | 0.1742 | 0.1429 | 0.1039 | 0.1039 | 0.1838 |
| * $DR_{BF-P0}^{QSGD}$ | 0.0190 | 0.0713 | 0.0198 | 0.0706 | 0.0852 | 0.0859 | 0.0621 | 0.0621 | 0.1836 |

This is evident in Figure 16 as SketchML compresses less aggressively than the other compressors, thus it achieves the highest accuracy. In contrast, 3LC sends the least data and the accuracy suffers the most. This is because 3LC is a hybrid method [84] that quantizes the values into $(-1, 0, 1)$ and selects the non-zero values for encoding. We use the default setting of 3LC that gives the least sparsification (but fails to recover the baseline accuracy). DeepReduce can be applied on top of any sparsifier and the sparsification ratio is flexible to choose. Figure 16 shows that DeepReduce on Top-$r$ not only reduces the data volume, but also improves accuracy by 0.7%.

**Simulated FL experiments in a bandwidth-limited local environment.** Apart from the realistic multi-region deployment, we also test DeepReduce with clients and the server in the same region connected with 100 Mbps network to simulate the low bandwidth scenario. The results are shown is Table 4. The total encoding/decoding overhead of DeepReduce is about 1.5-3.4× higher than Top-$r$. In contrast, the communication time is decreased by 1.8-4.0× for different DeepReduce variants, compared with Top-$r$. However, the extra overhead of DeepReduce is relatively low compared to their communication time reduction. Even with compression overhead taken into account, $DR_{BF-P0}^{QSGD}$ is 2.2× faster than Top-$r$ and 7.8× faster than the Baseline. Unlike the multi-region case which is suffering from the high latency network, the communication time of DeepReduce here is proportional to the transmitted data volume.

**FL training of MobileNet with DeepReduce.** We report the FL training of MobileNet [36] on CIFAR-10 [48] dataset by using 10 clients. The CIFAR-10 dataset is partitioned into totally 10 clients by Latent Dirichlet Allocation (LDA). The experiment follows the same training procedure as the standard FedAVG algorithm. We use 64 local batch size, 0.001 learning rate, 1 local epoch and ADAM [46] optimizer for the clients. We train MobileNet for 800 rounds and achieve 88.17% Top-1 accuracy for the baseline, which is consistent with the FedML benchmark [33]. We use Top-$r$ ($r$=10%) as the sparsifier to generate sparse tensors for DeepReduce, and compression is applied bidirectionally with error feedback. Figure 18 shows that Top-$r$ slightly affects the convergence rate compared with the baseline. Nonetheless, applying DeepReduce on Top-$r$ does not compromise the convergence behavior and the final accuracy while largely reducing the data volume (see Table 5).

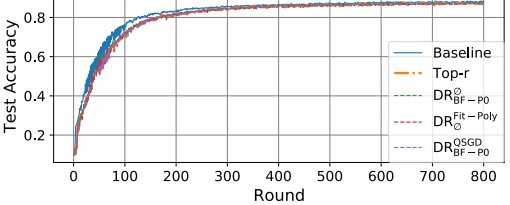

| Method | Relative Data Volume | Top1 Test Acc. |
|---|---|---|
| Baseline | 1.0 | 0.8817 |
| Top-$r$ | 0.2069 | 0.8708 |
| $DR_{\varnothing}^{Fit-Poly}$ | 0.1087 | 0.8700 |
| $DR_{BF-P0}^{\varnothing}$ | 0.1475 | 0.8758 |
| $DR_{BF-P0}^{QSGD}$ | 0.0713 | 0.8740 |

Figure 18: Convergence timeline of training MobileNet on CIFAR-10 datasets.

Table 5: Relative data volume and Top-1 test accuracy of MobileNet on CIFAR-10 in FL setup.

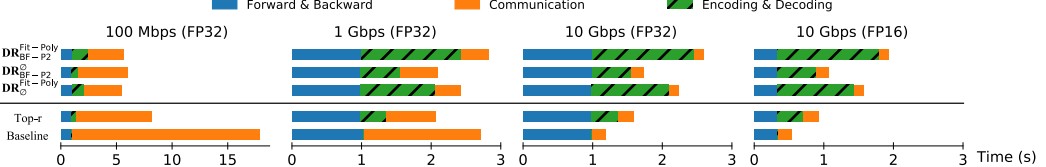

Figure 19: Time breakdown in one iteration training of NCF on ml-20m. We show the speedup of training by DR on 4 nodes with different network bandwidth: 100Mbps vs. 1Gbps vs. 10Gbps, and also with FP16 mixed precision training.

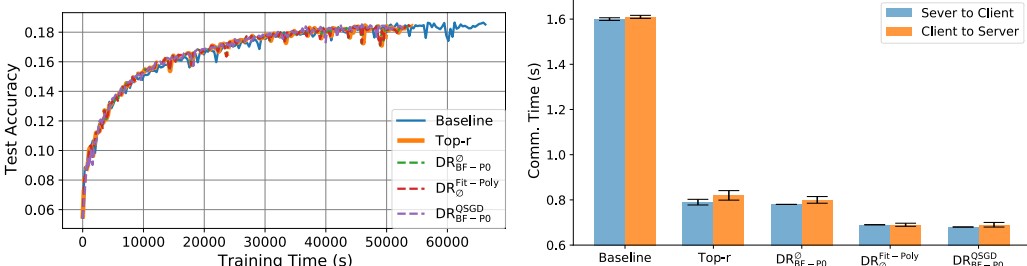

**Figure 20:** Convergence timeline of training an RNN to do next-word-prediction on Stack Overflow datasets.

**Figure 21:** Communication time (with error bars) for realistic Federated Learning deployments.

| Method | Relative Data Volume | Best Hit Rate |
|---|---|---|
| Baseline | 1.0 | 0.9497 |
| SKCompress | 0.2175 | 0.9513 |
| $DR_{BF-P0}^{QSGD}$ | 0.2063 | 0.9496 |

**Table 6:** We train NCF, an inherently sparse model, on ML-20m, with $10^6$ local batch size. We test $DR_{BF-P0}^{QSGD}$, which uses BF-P0 (FPR=0.6) for indices, but combines it with QSGD [7], an existing method for value compression. This demonstrates that DeepReduce is compatible with various existing compressors. We compare against SKCompress [40], an improved version of SketchML, optimized for sparse tensors. We configure QSGD and SKCompress for 7-bits quantization and set the QSGD bucket size to 512. For SKCompress, we omit the grouped MinMaxSketch and separation of positive/negative gradients, as they have only minor effects. All methods achieve virtually the same best hit rate (i.e., the quality metric for NCF), and both $DR_{BF-P0}^{QSGD}$ and SKCompress reduce the data volume by 5× compared with Baseline. However, in practice $DR_{BF-P0}^{QSGD}$ can be more easily implemented on GPUs. Therefore, in our experiments (see Figure 8b) it is 380× faster in terms of compression and decompression time.

---

**Algorithm 2** FedAvg with DeepReduce

---

**Input:** Number of clients $K$ indexed by $k$, local minibatch size $b$, number of local epochs $E$, learning rate $\eta$, DeepReduce compression:=DR, DeepReduce decompression:=DR$^{-1}$

**Output:** Trained model $x$

**On server side:**
Initialize $x_0$
**for** *round* $t = 1, 2, \ldots,$ **do**
   $S_t \leftarrow$ (random set of $m$ clients out of $K$ clients)
   $g_t \leftarrow DR(x_t - x_0)$
   **for** *each client* $k \in S_t$ *in parallel* **do**
      $g_{t+1}^k \leftarrow$ CLIENTUPDATE$(k, g_t)$
   $g_{t+1} \leftarrow \frac{1}{m}\sum_{k=1}^m DR^{-1}(g_{t+1}^k)$
   $x_{t+1} \leftarrow x_t - \eta g_{t+1}$

**On client** $k$ **side:**
Pull $x_0$ from server
**while** *training* **do**
   CLIENTUPDATE$(k, g_t)$

**function** CLIENTUPDATE$(k, g_t)$
Pull $g_t$ from server
$x_t \leftarrow x_0 + DR^{-1}(g_t)$
$g_{t+1} \leftarrow \mathbf{0}$
**for** *each local epoch* $i$ *from* 1 *to* $E$ **do**
   **for** *batch* $b \in$ *local training data* **do**
      $g_{t+1} \leftarrow g_{t+1} + \nabla\ell(x_t; b)$
Push DR$(g_{t+1})$ to server

---