# OpenReview forum: "DeepReduce: A Sparse-tensor Communication Framework for Federated Deep Learning"
_NeurIPS.cc/2021/Conference — NeurIPS 2021 Poster_

### Official Review · Reviewer_Rrbz · 2021-07-10

**Rating:** 7
**Confidence:** 4

**Summary:**

Problem: Communication overheads in the current FL setup, especially while transferring deep model gradients from clients to server and vice-versa.
Solution: The proposed DeepReduce framework offers to communicate sparse tensors in federated setup. Supports both the existing approaches for sparse communication as well as proposes two novel methods.
Results: A significant reduction in the transfer of data over the existing techniques.


**Ethical Concerns:**

None to the best of my knowledge

**Limitations And Societal Impact:**

The authors have to describe the limitations in detail.

However, my observation is that there is a significant room for improvement in overheads of computation, which will be useful for FL with meager resources on-device. The lossy compression with curve-fitting may need improvements

**Main Review:**

Distinction from state-of-the-art:
Well motivated with a focus on network bandwidths and communication costs. Accurately positions with-in the literature with a discussion on implicit ([18], [35]) and explicit ([6],[51],[68], etc) sparse models, different sparsification techniques. The entire related work section is also interesting.

Pros:
The paper is well written, easy to understand and digest the concepts.

The proposed method is solid with theoretical proofs and upper bounds on the errors for various parts such as curve fitting, etc.

The details of communication and compute (encoding and decoding times) values in Figure 8 are also impressive. Yes, there is room for improvement but this is an interesting ablation study.

Cons:

This is really a useful approach to save memory, especially very useful in low resource scenarios. However, the proposed approach as it is requires a few more additions for better understanding. Read the main+supplementary material, I can not see a standardization of the savings in space and communication costs along with the overheads in compute. To address these, i) please provide an overall algorithm, even in the appendix, if needed; ii) provide Big-O notations for the space, compute and communications; iii) compare and contrast with the existing methods. This actually gives a better understanding of how the proposed approach differs from the state-of-the-art.

The overhead of fitting an equation, V(i) for representing a sparse tensor sounds expensive. How accurate are these curve fittings? How many equations does one need per gradient? Does the error not propagate further with an approximation on these values?


The number of references are too many for a conference paper.


**Time Spent Reviewing:**

> 5

---

> ### Author Response · Authors · 2021-08-10
> **Response to Reviewer Rrbz**
>
> We are grateful to the reviewer for the constructive feedback and providing a positive assessment of our paper. We especially thank the reviewer for mentioning that the paper is “well-organized” and recognizing that the “proposed method is solid with theoretical proofs and upper bound.” The reviewer has raised some valid questions and provided many mindful suggestions. The following are our responses to the reviewer’s comments:
>
> **Overall Algorithm and Big-O notation, etc.** We thank the reviewer for this critical comment. We will surely provide: i) an overall algorithm in the Appendix and connect it with the main text for further clarification; ii) big-O notation for the space and compute complexity and compare with the existing methods.
>
> **How accurate are these curve fittings? How many equations does one need per gradient? Does the error not propagate further with an approximation on these values?**
> In the Appendix, we provided the approximation error of piece-wise constant fit, an existing result in [21], in Lemma 11. The approximation error of piece-wise linear fit with an explicit constant is a much stronger result and is quoted in Lemma 2 in the main paper. Both of these results provide the approximation error, $\|s-C_S(g)\|$ between the approximated vector, $s$ and the sorted vector, $C_S(g)$ and attest the accuracy of curve-fitting. While the approximation error of piece-wise constant fit depends linearly on the segmentation, $p$, the approximation error of piece-wise linear fit depends quadratically on $p$. That is, in theory, the approximation error in both cases approaches zero if $p\to\infty$. However, in practice, we use 5th degree polynomial (with segmentation) and double exponential (without segmentation) as fitting functions, and they lower the approximation error much efficiently compared to the piece-wise constant and linear fit. Figure 4 presents the accuracy of the fitting method on the convolution layer gradient of ResNet20 on the CIFAR-10 dataset by using eight segmentations---the black dotted curve is the sorted vector, $C_S(g)$ and the colorful segmentations are done via approximation vector, $s$.
>
> The number of equations per gradient equals the number of knots/segmentations we specify for the piece-wise fitting. The optimal number is determined by Lemma 11; please see the related discussion in Remark 4.
>
> The error from the curve fitting is given in Lemma 13 and the error from the combined index and value fitting is given in Lemma 14, in the Appendix. Although it is not clear what the reviewer meant by the *error propagation*, by using the standard techniques as in [23], based on Lemma 13 and 14, one can show the convergence of compressed distributed SGD for value-fitting and combined index and value fitting; please see Section D.1 in Appendix.
>
> **The number of references are too many for a conference paper.** We respectfully note that there is no absolute upper limit of citations specified by the NeuRIPS organizing committee. ML is an ever-evolving area of research, and hundreds of papers are coming out every day. We wanted our work to stay relevant in the perspective of these works. Therefore, we abundantly cited "relevant" works that fit the context of our paper. On the other hand, some reviewers can always consider "citing fewer works" as a pitfall of our work and can blame the credibility and scholarship of the authors. Therefore, we consider this comment as a non-issue.

---

### Official Review · Reviewer_E2i3 · 2021-07-16

**Rating:** 5
**Confidence:** 4

**Summary:**

This paper proposed framework DeepReduce for the compressed communication of sparse tensors. In practice, sparsed tensors will be represented by (Index, Value) set. DeepReduce contains a series of algorithm to further compress Indices and Values.

Firstly,  Bloom filter is a space-efficient probabilistic data structure, that is used to test whether an element is a member of a set. This paper use Bloom-filter as a lossy index compressor to determine whether an index is corresponding to a nonzero gradient. Furthermore, this paper proposes to use interpolation functions to fit a curve for relation between index and value.

**Limitations And Societal Impact:**

DeepReduce is a lossy gradient compression framework. Probabilistic and unpredictable errors is the biggest limitations. Compared with more stable compression algorithms, this framework does not show its superiority.

**Main Review:**

**Originality**:

This paper can be considered as a good combination of existing techniques.

**Quality**:

1. Bloom Filter is a random data structure with relatively high rate leading to false positives. Thus, some false positive indices may bring model divergence or unexpected outcome.
2. The gradients selected by Gradient Sparsification algorithm  tend to be important to model convergence in distributed learning. Using lossy compressor with numerical error is kind of waste in resource limited federated learning.
3. The trade-off between communication and precision is not well-explained.
4. Bloom-filter based index compressor is a important part in this paper, but the detailed implementation of Bloom-filter used in this paper is not clearly described (especially Hash functions used in Bloom-filter and actual hyper-parameters of Bloom-filter in experiments and so on).

**Clarity**:

This paper is well-organized but some points are not clear.  Some figures (e.g., figure 11, figure 12, algorithm 1) can improve reading experience, but can be found only in supplementary. The interpretation of mathematical symbols is quitemessy.

**Significance**:

The experimental evaluation is not sufficient.  The experiments in this paper did not show the advantages of its framework.

1. Top-r is not good enough to be a compression baseline. DGC  is widely accepted as a SOTA in Gradient compression. Gradient Quantization methods is also influential in gradient compression. But the paper does not have any comparative experiment with them.
2. From my perspective of view, index error and fitting error may cause model divergence, especially used in Gradient Sparsification (which means gradient selected by algorithm tend to be important).
3. In figure. 4, why  does the curve  of $DR_{BF-P2}^{\empty}$ and $ DR_{BF-P1}^{\empty}$ falls so much in 1000 - 2000 step?  Does it mean that BF compressor with great probability of loss information leading to model divergence.
4. In my opinion, there should be some experiments of $DR_{BF}^{Fit}$.



**Time Spent Reviewing:**

6

---

> ### Author Response · Authors · 2021-08-10
> **Response to Reviewer E2i3**
>
> We sincerely thank the reviewer for the effort in reviewing our paper. We respectfully mention that most issues raised by the reviewer are non-issues, and we would like to clarify the misunderstanding. Please see our responses below:
>
> **Effect of False Positives in Bloom Filter.**  The reviewer is correct and has made an important observation. We agree that the false positive (FP) responses in a Bloom filter (BF) can lead to misalignment of the gradient components, and that can potentially cause model divergence; please see our discussion on this in Lines 139-145. However, the paper’s novelty is that we do not use this naive approach; our BF is aware of the false-positive occurrences. In our case, the transmitting worker can execute the BF algorithm and determine the set of positives, $P$, before any communication. With that information, it constructs the array so that, even if an FP index is selected, the decoding device will get the gradient element corresponding to that FP index, avoiding misalignment. However, it has a tradeoff in terms of compression error; please see the theoretical results and discussions in Lemma 6, 7, 8, and 9. Also, please see Figure 4, where $DR_{BF-naive}^{\emptyset}$ behaves as you mentioned and has the slowest convergence, but DeepReduce with different false-positive aware BF policies perform better.
>
> **Compression in the FL algorithm is a kind of waste.** We respectfully disagree. The clients in Federated Learning (FL) are resource-constrained, geographically remote, and connected with insufficient or inadequate network bandwidth. As a result, communication is expensive, and hence compression is applied to remedy the network bottleneck. This has been reported in many works that use lossy compression in FL set-up, and this is a new de-facto norm in FL training, e.g., please see [30,31,60]. Therefore, we disagree that "Using lossy compressor with numerical error is kind of waste in resource-limited federated learning."
>
> **The trade-off between communication and precision is not well-explained.** We respectfully disagree. On the contrary, we explained it in detail in the main paper, with sufficiently cross-referencing the formal Lemmas and proofs in the Appendix. Please see Lemma 6 in the Appendix (quoted in the main paper, Lines 153-154). This Lemma explains how much extra data volume is used while policy P0, with different false positive rate (FPR), $\epsilon>0$, is used to communicate a $r$-sparse gradient vector. Complementing this result, Lemma 7 explains how the compression error (which, in other words, is equivalent to the precision) gets affected. The upside of sending extra data via P0 is expected to lower the compression error, and that is precisely the message of Lemma 7---please see Remark 3 and Lines 709-714 in Appendix. When we use other policies, say P1, the data volume reduces than P0. However, the compression error is affected and is given in Lemma 9 in Appendix. The excerpt of this result was mentioned in the main paper; please see Lines 160-165. Finally, the error for polynomial fit was given in Lemma 2 of the main paper, see Line 212, and a dedicated, detailed discussion and proofs are provided in Appendix. Last but not least---We provided a theoretical analysis of compression error from value compression in Appendix C.1; the compression error from joint value and index compression in Appendix D; and provide comments regarding the convergence of these approaches in Appendix D.1.
>
> **Hash-Functions and Bloom Filter Implementation.** We thank the reviewer for this important suggestion. We will add the implementation details. We use MurmurHash (MurmurHash3) to construct the hash table in the GPU implementation (Python library link: https://pypi.org/project/mmh3/) of the Bloom filter. We will clarify the procedure of determining the number of hash functions, $k$, and the length of Bloom filter bit-string, $m$, which is already described in Lemma 3 and Remark 2 in the Appendix. The only hyperparameter in the BF experiments is the FPR, indicated in the related captions; please see Figure 4,14, Table 6, and the related texts (Line numbers 246, 264, and 987).
>
> **Insufficient experiments---DGC and Gradient quantization technique.** We respectfully disagree with the reviewer. DGC [51] is nothing but the Top-r compression technique. In DGC, based on a user-defined threshold, $\gamma>0$, the Top-$r$ gradient components are selected---Any gradient component with magnitude greater than or equal to $\gamma$ is kept, and others are set to zero---mimicking the same sparsification technique as Top-$r$; please see Lines 9-12 in Algorithm 1 in DGC [51]. On the other hand, we compared with one of the most widely-adopted, influential, and state-of-the-art gradient quantization techniques, QSGD [7], for our value-compression---Please see Table 2, Figure 6 (in FL set-up) and Figure 8 (for data-center set-up) in the main paper, and Figure 18 (in FL set-up), Table 4-6 in Appendix.
>
> **Index error and fitting error may cause model divergence.** This is a valid point, and we thank the reviewer for that. Our short answer is “No.” Please allow us to clarify. We will not incur any index-compression error due to the "misallocation" of the indices in the Bloom-Filter-based index compression. Please see our comments in response to the *Quality* of the paper. In continuation to that comment, however bad the policy performs in index selection, the compression error of the resulting sparse gradient in expectation will be no worse than using a Random-$r$ sparsifier on the original gradient; please see Lemmas 7-9. We know that compressed, distributed SGD with $\delta$-sparsifier, such as Random-$r$, retains the same asymptotic convergence of baseline, no-compression SGD (cf. [68, 23]). Therefore, index fitting with BF and using different policies will not cause model divergence; please see our convergence comment in Appendix D.1. Similarly, the compression error from value compression is given in Appendix C.1. The combined compression error from index and value compression is given in Appendix D. With the above, based on the strong growth condition of stochastic gradients [23,76], for a lower bounded, Lipschitz smooth, non-convex loss function, $f$, following [23], distributed SGD with any $\delta$-sparsifier after using an individual index or value compression, or both converges. Therefore, the reviewer’s perspective "index error and fitting error may cause model divergence" is incorrect.
>
> **In figure. 4, why does the curve of DR(BF-P2) and DR(BF-P1) falls so much in the 1000 - 2000 step? Does it mean that BF compressor with great probability of loss information leading to model divergence?**
> Although all our bloom policies are *FP-aware*, their capability of maintaining the original information varies. P0 is lossless, while P1 and P2 may suffer from information loss and cause slow convergence but never lead to divergence (please see our theoretical guarantees Lemma 6-9 in Appendix). As we explained in the previous comment, the compression error of the resulting sparse gradient from P1 in expectation will be no worse than using a Random-$r$ sparsifier on the original gradient. Policy, P2 being a general P1, will also incur a similar error. Indeed, policies P1 and P2 are worse than the original sparsifier and policy P0. Their benefit lies in controlling the data volume, and their convergence behavior is the same as the Random-$r$ sparsifier. A well-known result states that distributed SGD with Random-$r$ sparsifier converges at the same asymptotic rate as the no-compression baseline SGD [23,76]. Therefore, although it might appear that “curves of DR(BF-P2) and DR(BF-P1) fall so much in the 1000 - 2000 step,” at the end of the training, they will attain the baseline test accuracy, as is evident from Figure 4.  Moreover, in  Figure 4, we apply the BF compressor on top of Top-$r$, where $r$ is set to 0.01. This extremely low compression ratio may make the model highly sensitive to information loss at the beginning but does not affect the final test accuracy.
>
> **Experiments on DR(Fit-BF).** We respectfully request the reviewer see the results in Figure 7 for DR(Fit-BF) as suggested.

---

> > ### Comment · Reviewer_E2i3 · 2021-08-30
> > **Thanks for the detailed response**
> >
> > Thanks for the detailed response to my questions.  It is helpful for me to better understand this work.
> >
> >   1. The response feedback did not address my concern on the experimental results on $DR_{BF}^{Fit}$​ . There are  no experimental results showing how the error rate of $DR_{BF}^{Fit}$​ converge. There are only results only $DR_{Fit}^\empty$  in Figure 4.
> >
> >   2. The top-r gradients are important to model precision. Sparsifying these top-r gradients with bloom filter may bring high false-positive, which may lead to further losses of important gradients with slow convergence. Therefore, disregarding the important gradients(which however should be better utilized) is a kind of waste in the resource-limited setting. The reviewer does not mean lossy compressor with numerical error is not acceptable. Instead, further explantion on how the discarded gradients impair the convergence is required. Especially, as shown in Figure 4, the proposed algorithms degenerated a lot after about 1500 steps.   The feedback does not answer this question. Further, did these algorithms perform similarly on larger datasets with more classes, e.g., imageNet or CIFAR-100 with more classes?
> >
> >   3. The reviewer agrees that DGC is a  top-r approach. However, it is better to list the results of DGC for further demonstating the wide applicablity of the proposed method.
> >
> > In summary, the major contribution lies in the improvement of communication compression performance in communication-limited setting. However, the limitations should also be clarified in the paper. First, the existence of false positive samples limits its usage in high-precision settings; second, for data-center federated learning where there is usually sufficient communication resources, compressing the gradients using DR may lead to lower precision with high risk of slow convergence.

---

> > > ### Author Response · Authors · 2021-08-30
> > > **Response to the reviewer's queries**
> > >
> > > We sincerely thank the reviewer for the time taken to read our rebuttal. We are glad that our rebuttal helped the reviewer understanding our work better. However, there are still some misunderstandings that the reviewer pointed out. We would like to humbly mention that most of these results already existed in the paper. We understand that the reviewer might have missed them due to tremendous reviewing pressure. Below we address the key points mentioned by the reviewer:
> > >
> > > **$DR_{BF}^{Fit-Poly}$ in Figure 7.** We respectfully disagree with the reviewer. There is an experimental result on $DR_{BF}^{Fit-Poly}$ in Figure 7 that shows the **Time breakdown in one iteration training of NCF on ml-20m. We show the speedup of training with different network bandwidths.** Moreover, we provide a more detailed result of the same experiment with more diverse network bandwidths; please see Figure 19 in the Appendix. Therefore, the reviewer’s claim is incorrect, and we request the reviewer to recheck the paper kindly. We also respectfully note that in the first review, the reviewer said, “In my opinion, there should be some experiments of $DR_{BF}^{Fit-Poly}$” and we responded to that. However, now the reviewer asserts, “There are no experimental results showing how the error rate of DRBFFit converges.” Admittedly, it is always possible to do more experiments, but one should not overwhelm the reader. Therefore, we do not feel it is necessary to show the convergence behavior of $DR_{BF}^{Fit-Poly}$ further as we have many convergence plots that show the effectiveness and convergence of our proposed techniques.
> > >
> > > **Disregarding the significant gradient components (which should be better utilized) is a kind of waste in the resource-limited setting.** The reviewer is undoubtedly not correct with his claim, and we respectfully disagree with the reviewer unless a more robust logic is provided to point out the claim is correct. We respectfully reiterate, however bad any BF policy performs in index selection, the resulting sparse gradient compression error in expectation will be no worse than using a Random-$r$ sparsifier on the original gradient; please see Lemmas 7-9. A well-known result states that distributed SGD with Random-$r$ sparsifier converges at the same asymptotic rate as the no-compression baseline SGD [23,76]. Therefore, unless the reviewer points out significant discrepancies in our theoretical results, Lemmas 7-9, and the convergence theorems in [23,76], we can not simply accept the comment that “Sparsifying these top-r gradients with bloom filter may bring high false-positive, which may lead to further losses of important gradients with slow convergence.” For convergence of our proposed methods, we provided sound mathematical logic, and we humbly mention that we do not want to entertain a heuristic argument to negate mathematical logic. The reviewer is welcome to find flaws in our results and destroy them, but we kindly request not to find flaws without proper logical reasoning.
> > >
> > > **Experimental results on larger datasets with more classes---ImageNet classification.** The reviewer asked about the performance of the DeepReduce algorithms on larger datasets. We respectfully mention that **we already provided results** of the DeepReduce on ResNet-50 benchmark on ImageNet dataset comparing against Top-r, SketchML with 64 states, 3LC, and baseline no compression SGD. We also mentioned these experiments in the main paper; please see **Lines 271-272, where we wrote, “In Appendix F.3, we also compare DeepReduce against 3LC [50] and SketchML [39] on a larger benchmark, ResNet-50 on ImageNet”.** Therefore, we request the reviewer to reread our paper carefully and check the results kindly. **Importantly, Figure 16 shows that DeepReduce on top of the Top-r reduces the data volume and improves accuracy by 0.7%, a significant gain on the ImageNet task.** Therefore, we respectfully mention that the reviewer is wrong, claiming, “Sparsifying these top-r gradients with bloom filter may bring high false-positive, which may lead to further losses of important gradients with slow convergence.”
> > >
> > > **Result on DGC.** This paper compared DeepReduce against five sparsification strategies, Top-$r$, Random-$r$, SketchML, SketchCompress, and 3LC; no compression baseline SGD and FedAvg to show our results. Moreover, we integrated naive BF and BF with three policies (P0, P1, P2), RLE, Huffman, and Deflate in DeepReduce index compression, QSGD, double exponential fitting, and polynomial fitting in DeepReduce value compression; please see Table 3, *Summary of implementation*. We have already put tremendous effort in carefully comparing and presenting our result in the scope of a nine pages conference paper. Therefore, if the reviewer agrees that DGC is a Top-$r$ approach, there is no point in further demonstrating the results of DGC---DeepReduce on top of the Top-$r$ sparsifier already performs better; please see Figures 4-5 (Test accuracy in an end to result), Figure 8 (a) (relative data-volume), Figures 15-17 in the Appendix (Top 1 accuracy vs. data volume), Tables 2 and 5 (Top 1 accuracy vs. data volume in FL setup). However, as the reviewer mentioned, we can discuss the applicability of DGC in DeepReduce in the revised version.
> > >
> > > Taken together, we humbly mention that the issues raised by the reviewer are either non-issues or were already presented in the manuscript. Therefore, we request the reviewer to recheck the paper carefully and reevaluate their assessment in light of our response and the other reviewers' positive feedback on the paper.

---

> > > > ### Comment · Reviewer_E2i3 · 2021-09-04
> > > > **There is contributions in this paper, but concerns need be addressed**
> > > >
> > > > The reviewer regards this paper as of an interesting trade-off research between communication and computation. As shown in figure 8,  the proposed methods spend less time to compress but cost more data volume compared with SKCompress. However, the following issues has not been addressed in a convincing way.
> > > >
> > > > This paper proposed P0, P1 and P2, three different bloom-filter based compressor for indices and three value fitting strategies for values compression. I agree that some interesting contributions has been made in this paper.  Varies of experiment results indicates the compression strategies are useful for different experiment setting.
> > > >
> > > > Thanks for reminding the appendix. Sorry that duing  the response stage, the appendix is not available in the system. However,  I do find some issues to improve the quality of this paper. Comparing  figure 16-17, the policy used in two datasets are different.  For consistentcy, please report results of the same policies on these data sets.
> > > >
> > > > The experimental results on $DR_{BF}^{Fit-Poly}$ in figure 7 and figure 19 only shows the speedup of training, but no experimental results show its effect on model performance since this paper claims that compressors in this paper do not affect the quality of the model. However experiments results of $DR_{BF-P1}$ and $DR_{BF-P2}$ in figure 4 shows the limitations of BF-P1 and Bf-P2.  And no persuasive explanations were given on why the curve falls so much in 1000-2000 steps and stay little improved in 2000-2800 steps. Therefore, it is reasonable to request  the authors add a convergence curve of $DR_{BF}^{Fit}$ .
> > > >
> > > > Two effective compressor with different strategies are proposed in this paper, and it is reasonable to see a powerful combination of them. That's why I insist on adding more detailed experimental results of $DR_{BF}^{Fit}$ to prove the effectiveness.
> > > >
> > > > DGC(published in 2017) is a related variant and out-performs top-r.   A high quality NeurIPS paper should have a convincing comparison with recent state-of-the-art results (especially those published in the past two years).
> > > >
> > > > I noticed another problem from Reviewer *j5Bs*, which is "16MB model with 200 training round leads to 3200MB memory consumption".  I also worries about its usage on realistic DL tasks, which may cause expensive memory consumption.
> > > >
> > > > To sum up, I appreciate the author's active response to my questions and I do agree the interesting contributions made in this paper. But the reviewer is not convinced that this paper is a high-quality NeurIPS paper.

---

### Official Review · Reviewer_qVgi · 2021-07-19

**Rating:** 7
**Confidence:** 3

**Summary:**

The work proposes a general framework for sparse gradient transfer in distributed deep learning, where the non-zero indices and values are considered and compressed separately. In addition, authors propose new mechanisms for both index and value compression, demonstrating their advantages both theoretically and empirically.

**Limitations And Societal Impact:**

Yes

**Main Review:**

Strengths:
1) The work is well-written and easy to understand: specifically, the illustrations help the reader gain enough insight into the studied methods and their performance.
2) Both proposed compression methods are simple to implement and have theoretical guarantees while exhibiting favorable performance when compared to the baselines.
3) Authors provide a general framework for sparse data transfer, which might be used as a foundation for future methods.

Weaknesses:
1) The work cites recent works on gradient compression (for instance, via low-rank decomposition, such as PowerSGD), yet in the experiments, there are no comparisons with these methods — only the sparsity-based ones. To clarify the applicability and relative performance of DeepReduce, it might be useful to see a comparison with a broader range of communication-efficient distributed methods.
2) Although Table 2 and Figure 7 provide time breakdowns for the methods, it might be useful to plot the convergence curves of Figure 6 with wall clock time on the X-axis. In addition, one important detail missing from the experimental setup is the distribution of client-server bandwidth, which can help the reader gain insight into the specifics of the studied setting.
3) Despite the indication in the checklist, I could not find the error bars for the experiments. They would be especially helpful in Table 2, because decoding and communication times are quite similar for the compared versions of DeepReduce.

**Time Spent Reviewing:**

4

---

> ### Author Response · Authors · 2021-08-10
> **Response to Reviewer qVgi**
>
> We sincerely thank the reviewer for the positive assessment of our paper and constructive feedback. The reviewer has correctly identified that our framework “might be used as a foundation for future methods.” Below we address the key points mentioned by the reviewer:
>
> **Comparison with PowerSGD.** ​​PowerSGD [by Vogels et al. NeuRIPS, 2019] or any low-rank approximation methods use for gradient compression, decompose the original gradient matrix, $M$ (where gradient vector from different layers of the DNN are stacked as the columns of $M$) into two low-rank matrices, $P$ and $Q$, say. However, in the DeepReduce framework, we treat the gradients from each layer of a DNN as a rank-1 tensor (a vector) and use our index and value compression on them. Therefore, PowerSGD is a distant approach to us. Moreover, DeepReduce is particularly tailored for sparse vectors, either from a sparsifier [68,70,75] or a direct artifact of inherently sparse DNN gradients [18,35]. Therefore, we compared DeepReduce against state-of-the-art sparse gradient compression techniques (see Figure 8) and did not compare with low-rank decomposition techniques, such as Power-SGD, as it is not coherent. After this, hopefully, the reviewer agrees that it is not a weakness of this work.
>
>
> **Convergence Curves and Error Bars.** We thank the reviewer for these important suggestions. Indeed, we will plot the convergence curves of Figure 6 with wall clock time on the X-axis and the error bars in the main paper. Since we can not attach any figure or real table in the scope of this rebuttal, please see our sample result below. In the *Average Communication Time* column, the first column denotes the server to client communication time, and the second column denotes the client to server communication time.
>
> ============================================================================================
>
> Method            |             Average Communication Time (In second)       |       Training time (In minute)          |    Accuracy
>
> ============================================================================================
>
> Baseline           |            $1.60\pm0.0059$     |   $1.61\pm0.0066$           |          1109.7                                   |  0.1856
>
> ------------------------------------------------------------------------------------------------------------------------------------------------------------------
>
> Top-$r$ (10 \%)   |       $0.79 \pm 0.0125$   |   $0.82\pm 0.0211$          |              907.6                                  |  0.184
>
> ------------------------------------------------------------------------------------------------------------------------------------------------------------------
>
> ${DR}_{{BF}-{P} 0}^{\varnothing}$ |   $0.78 \pm 0.0002$ | $0.8 \pm 0.0148$ |    902.7                                    | 0.1841
>
> ------------------------------------------------------------------------------------------------------------------------------------------------------------------
> ${DR}_{\varnothing}^{{Fit-Poly }}$| $0.69 \pm 0.0002$ | $0.69 \pm 0.0068$  | 902.8                                         | 0.1838
>
> ------------------------------------------------------------------------------------------------------------------------------------------------------------------
> ${DR}_{{BF}-\mathrm{P} 0}^{{QSGD}}$ | $0.68 \pm 0.0002$ | $0.69 \pm 0.0102$  |  892.6                             | 0.1836
>
> ------------------------------------------------------------------------------------------------------------------------------------------------------------------

---

> > ### Comment · Reviewer_qVgi · 2021-08-26
> > **Thank you for the response!**
> >
> > Thank you for a detailed response to my concerns! After carefully reevaluating the work in light of the discussion between authors and all reviewers and the new results, I have decided to increase its score from 6 to 7.

---

### Official Review · Reviewer_j5Bs · 2021-07-21

**Rating:** 5
**Confidence:** 4

**Summary:**

This paper proposed a framework to compress sparse tensors in the context of federated learning, in order to further save the amount of transmitted data between the server and clients. The proposed method could be incorporated into the existing pipeline of distributed training with sparsification. The main idea is to compress/decompress the index and value separately with different compression methods. The paper comes up with two compression methods, i.e., Bloom-filter based index compressor and curve-fitting based value compressor. The paper shows some synthesized and realistic experiments to justify the proposed methods.

**Limitations And Societal Impact:**

The discussions about limitations are not enough. For example, the proposed framework is open to use different compressors and combinations for index and value. And we can see these combinations have very different behaviors. How to choose these combinations to get a good balance between accuracy and efficiency is not discussed.

Besides, data privacy is one of the most important motivations of federated learning. However, the author didn't discuss the potential impact on data privacy. Since this is a framework specifically intended for FL, I think it's necessary to take data privacy into account.

**Main Review:**

This paper is targeting a meaningful problem, that is, minimizing the communication data amount in the distributed training, especially federated learning where the training devices (like mobile phones) are assumed to be massively distributed and could be charged by the data traffic. Also, the paper did a detailed analysis (in both theory and experiments) about compression methods of sparse tensors.  It is interesting to see the behavior of different combinations of compression methods on sparse tensors (Fig 4).

However, I do have several concerns.
1. As a FL algorithm, it is not clear that how exactly the training process works on server and client respectively. As we know, the distributed learning process can be very specific, e.g., what do you communicate, model or gradient? when do you communicate? how to update the model? and so on. If you are following the FedAvg stereotype where clients receive model parameters from the server and send gradients back to the server, then how do you apply your method on model parameters which are usually not sparse? If not, then you may have a different training process than FedAvg, which is not specified in the paper.

2. As shown in Fig8, the SKcompression actually has lower data volume than the proposed method, but it takes more time for encoding. Therefore, we still don't know if the proposed method can be faster in the end-to-end comparison with SKcompression. I guess it depends on this is computation or communication that dominates the overall time. Maybe adding SKcompression result in Fig 7 can make it more clear.

3. The experiment setting is a bit confusing for me. It seems you used 56 AWS instances in total, but in 6.2, the paper mentioned the dataset is partitioned among 342,477 clients, and the server randomly selects 56 clients to communicate. So I assume each instance has multiple clients? This part needs more descriptions.

4. In Table 1, the optimizer column is a bit confusing. FedAvg is a distributed training algorithm, while SGD and Adam are optimization methods. I don't know how to compare these two concepts.

5. The paper mentioned multiple times the "easy-to-use API" of the proposed framework, and has a section to describe the "system architecture". These may give the audience an expectation that this work is building up a system. But we didn't see the detailed information about the API and the system. Also, the limitation of the system is not discussed.

**Time Spent Reviewing:**

4

---

> ### Author Response · Authors · 2021-08-10
> **Response to Reviewer j5Bs**
>
> We are thankful to the reviewer for reviewing our paper, providing many mindful suggestions, and noting that the paper did a “detailed analysis (in both theory and experiments).” We address the comments of the reviewer as follows:
>
> **Training Process in FL and the details.** We thank the reviewer for pointing this out. We have noticed that the model parameters are usually not sparse. Thus we apply compression on the model updates instead---the model difference between the current and last steps. This is eventually the scaled gradient at each node. Without compression, pulling model updates is equivalent to pulling model parameters. We will clarify this in the main paper.
>
> **Experimental Set-up on AWS, SKCompress result.** In FL, it is usually not applicable to conduct an actual experiment on many devices where each device takes only one data partition. Thus, we follow the popular simulation approach: each physical device takes multiple data partitions to represent multiple clients. We will clarify this in the final version. Additionally, as the reviewer suggested, we will add the SKCompress result in Figure 7 to make it clearer.
>
> **API and System-Architecture, Limitations, and data-privacy.** We thank the reviewer for mentioning the API and system architecture. As the reviewer suggested, we will add the API description; it resides on our anonymous GITHUB page. Please see Lines 1038-1040 in the Appendix for a link to the anonymous GITHUB page.
>
> DeepReduce may suffer from slower training if the network bandwidth is extremely high. This may limit its use for the data-center set-up, where the network bandwidth is generally way higher than the FL set-up. Therefore, we use DeepReduce as a sparse tensor communication framework for FL. We will add this limitation in the existing Section 6.3, *Practical applicability of DeepReduce*.
>
> Next, we thank the reviewer for asking us to discuss the impact of data privacy in the FL setting. Indeed we will provide that.
>
> **FedAvg vs. SGD and ADAM.** We thank the reviewer for this comment. To clarify: FedAvg is a distributed training algorithm used for federated learning, with SGD as a local optimizer for each client/worker. On the other hand, there is no algorithm with such a name in a distributed data-center setup. Therefore, we use SGD and ADAM as optimization algorithms for each client in a distributed data-center setup. We will clarify this in the paper.
>
> **Limitations: Combinations of Compressors.** The reviewer has correctly asserted that different compressors "have very different behaviors.” This was also observed in a concurrent survey and compression framework by Xu et al. [83]. Nevertheless, how to choose a better combination of compressors is a complicated research problem on its own. Because no one compressor outperforms others, their performance differs depending on the DNN architecture, network bandwidth, and other constraints; please see this observation in [83]. Our goal in this paper was to decompose the sparse tensor into two sets, indices and values, and then propose a framework that allows for independent and combined compression of these two sets by using existing and our compression strategies. We respectfully mention, doing a comparison study with the combinations of different compressors is not in the scope of this paper but is a good direction for future work.

---

> > ### Comment · Reviewer_j5Bs · 2021-08-24
> > **response to the rebuttal**
> >
> > Thanks for the authors' kind reply regarding my questions, which is quite helpful for me to better understand your work. However, I do still have two specific questions following up your reply.
> >
> > The first one is about the efficiency of the proposed method. The trade-off between computation and communication is a common issue for communication compression training algorithms. From this perspective, we can see that the proposed method has a longer computation time but shorter communication time than top-r, which leads to a faster overall time than top-r in the low bandwidth situation. Since there is no such comparison with the SKcompress, which is more like the SOTA method, it is not clear that in which way the proposed method can outperform the SKcompress. My guess is that the proposed method is somewhere between top-r and SKcompress, that is, it has faster communication than top-r but faster computation than SKcompress. It could be an interesting trade-off study here.
> >
> > The second one is about the details of the algorithm. I agree that compressing model differences is a common strategy in compressed training. But it is more complicated in the context of FL. For example, when a selected worker needs to pull the latest model from the server, it actually pulls the difference between the server model and its own local model. Then I assume the server needs to know the local models of all workers? because the workers could participate in different communication rounds and end up with different models in FL. I am curious how do you deal with this issue.

---

> > > ### Author Response · Authors · 2021-08-26
> > > **Response to the reviewer's queries**
> > >
> > > We sincerely appreciate the reviewer’s effort in reading our rebuttal and asking us two interesting and relevant questions. Please see our response below.
> > >
> > > **SKCompress.** We agree that SKCompress [40] is the state-of-the-art method for compressing sparse tensors. However, it has a different targeting scenario compared to our proposed methods. SKCompress is built originally for simpler ML models such as logistic regression, support vector machine, and linear regression (please see Table 2 and Figures 10-12 in [40], except it also trains ResNet18 model on Cifar10 dataset), where the training is usually conducted on CPU clusters. In contrast, our proposed methods are exclusively tailored for federated deep learning, where GPUs are usually used for acceleration. The essential advantage of our proposed methods is the easy implementation and the efficient execution on GPU, which enables faster training via sparse tensor compression in the FL training. In Figure 8 (b), we already show that SKCompress is *two orders of magnitude slower* than our proposed methods. In the actual end-to-end experiment, SKCompress is about *43 times slower than the baseline (~18 hours)*, which makes it impractical to finish the whole training job. Jiang et al. showed in [40] that SKCompress has *five times more significant computation overhead* than communication in the DNN training on GPUs. Thus Jiang et al. in [40] rightfully concluded that “SKCompress is not suitable for deep learning tasks”---Please see Section 7.7 Limitation in [40].
> > >
> > > Regrading the communication efficiency, which the data volume can directly reflect in a constant bandwidth network, SKCompress outperforms all the competitors in the benchmark, as shown in Figure 8 (a). However, in some cases, DeepReduce can still have a minor advantage over SKCompress; please see Table 6 in the Appendix. We understand and appreciate that this is a valid point in the paper’s context, and we will clarify this in the revised version.
> > >
> > > **Details of the Algorithm.** We again thank the reviewer for this relevant question. Indeed, the central server needs to keep all previous model states during the training phase of our algorithm. When a client asks for the latest model, it also reveals the current model state to the server. In our FL experiment, the model size is about 16 MB, and the total training round is 200, which means the total space cost is approximately 3200 MB. We will clarify this in the main paper.
> > >
> > > In light of the above response, we sincerely request the reviewer to reevaluate our paper.

---

> > > > ### Comment · Reviewer_j5Bs · 2021-08-30
> > > > **following up discussion**
> > > >
> > > > Thanks for your explanations regarding my questions. Overall, I do agree that this paper did some interesting theoretical work regarding sparsification, but I can hardly be convinced that this work is a general federated learning framework. The author emphasized that this work is intended for GPU-based DNN tasks, which is also one of its novelties compared with the SOAT, but apparently, the GPU memory cost of the proposed method increases linearly with the model size and communication rounds. The model size reported in the experiment (16MB) is way too small in the context of compressed distributed training. I am afraid that for more realistic DNN tasks (usually hundreds of MB or a few GB of model size, hundreds of communication rounds), the memory cost of the proposed method will be prohibitively expensive for any GPU device.

---

> > > > > ### Author Response · Authors · 2021-08-31
> > > > > **Response to follow-up discussion**
> > > > >
> > > > > We thank the reviewer to appreciate our work again. We respectfully disagree with the reviewer that our work is not applicable to FL. We note that the memory being discussed here is the server memory, and in FL, the server only performs client update aggregation. In the traditional FL, the central server is usually a CPU-based system. While in the deep FL context, GPUs are used to accelerate the computation process, such as the local DNN training and the compression/decompression process. Since no training happens on the server, the server does not need to have a GPU, and using memory limited GPUs for storing huge model states is a waste. Therefore, this server can be assumed to have significantly better storage and computational capabilities in comparison to the client devices. In our implementation, we keep the frequently accessed model states (usually within a certain number of communication rounds) in the host memory and persist the infrequent ones on the local SSD. As a result, as the reviewer mentioned, storing larger model states on the resource-abundant server-side can be easily handled. There is no need to store model states on the clients. Additionally, we mention, the FL benchmark we used here is the SOTA next work-prediction model proposed by the Google team [59]. We envision that with larger models the speedup would be much more significant.

---

### Comment · Area_Chair_eYsr · 2021-08-19
**Is it a generic tensor compression algorithm or a specific algorithm for FDL?**

To me, this paper proposes a generic tensor compression algorithm other than any specific algorithm for FDL.

Can authors of this paper clarify of it?

---

> ### Author Response · Authors · 2021-08-20
> **Gains in FDL are more significant in practice**
>
> Indeed, our algorithm is generic and agnostic of the application.
> However, our sparse representation is lossy; therefore, the application should be able to tolerate this.
> Currently, we are targeting ML training exactly because it is tolerant to the lossy representation; however, we do not exclude other applications.
> We are specifically targeting Federated Distributed Learning because the practical gains from the compressed representation are significant, due to the limited network bandwidth. Our method can also be used for training within a single data center, but in this case the gains are quite limited in practice, because of the fast communication (typically more than 100Gbps).

---

### Decision · Program_Chairs · 2021-09-27

**Decision:**

Accept (Poster)

**Comment:**

This paper proposes a compression algorithm to compress the tensor communicated in the FL application to reduce communication cost, with certain theoretical guarantee for the compression. Reviewers generally agree that there do exist novelty for this compression method.

All reviewers read the rebuttal. Reviewers (especially the ones with concerns) made serious 2nd or 3rd response to the rebuttal and participated the discussion seriously. Based on scores, this is really a borderline paper. We are trying to identify the real contribution of this paper to be 1) a new FL algorithm or 2) a new compression algorithm. This paper is somehow between 1) and 2). We suggest authors to revise this work to make one single point solid enough by appropriate experimental design, if this paper is accepted.

If this paper is considered to sell a new communication efficient FL algorithm, the comparison to other recent FL algorithms should be comprehensively compared. For example, the tradeoff between computation and communication can be more clear; The memory issue (it requires a large memory to store historical models for communications efficient) needs to be treated and discussed seriously, as pointed out by a reviewer; How is the accuracy affected by the compression?

The proposed key compression algorithm is quite general. It can be used to many different areas. FL is just one of them. If this paper's main contribution is considered to be a new compression algorithm, then the comparison to other compression algorithms need to be compared more carefully. The application to other fields need to be shown.